# The evolutionary mechanism of non-carbapenemase carbapenem-resistant phenotypes in *Klebsiella* spp

Natalia C Rosas[1,2†], Jonathan Wilksch[1,2†], Jake Barber[1,3†], Jiahui Li[1,2,4], Yanan Wang[1,2,3,5], Zhewei Sun[6], Andrea Rocker[2], Chaille T Webb[1,2], Laura Perlaza-Jiménez[1,2], Christopher J Stubenrauch[1,2], Vijaykrishna Dhanasekaran[1,7], Jiangning Song[1,4], George Taiaroa[8], Mark Davies[8], Richard A Strugnell[8], Qiyu Bao[6], Tieli Zhou[4]*, Michael J McDonald[1,3]*, Trevor Lithgow[1,2]*

[1]Centre to Impact AMR, Monash University, Clayton, Australia; [2]Infection Program, Biomedicine Discovery Institute and Department of Microbiology, Monash University, Clayton, Australia; [3]School of Biological Sciences, Monash University, Clayton, Australia; [4]The First Affiliated Hospital of Wenzhou Medical University, Wenzhou, China; [5]Infection Program, Biomedicine Discovery Institute and Department of Biochemistry & Molecular Biology, Monash University, Clayton, Victoria, Australia; [6]Wenzhou Medical University, Wenzhou, China; [7]School of Public Health, LKS Faculty of Medicine, The University of Hong Kong, Hong Kong Special Administrative Region, China; [8]Department of Microbiology and Immunology, The Peter Doherty Institute, The University of Melbourne, Melbourne, Australia

**\*For correspondence:**
wyztli@163.com (TZ);
mike.mcdonald@monash.edu (MJM);
trevor.lithgow@monash.edu.au (TL)

†These authors contributed equally to this work

**Competing interest:** The authors declare that no competing interests exist.

**Abstract** Antibiotic resistance is driven by selection, but the degree to which a bacterial strain's evolutionary history shapes the mechanism and strength of resistance remains an open question. Here, we reconstruct the genetic and evolutionary mechanisms of carbapenem resistance in a clinical isolate of *Klebsiella quasipneumoniae*. A combination of short- and long-read sequencing, machine learning, and genetic and enzymatic analyses established that this carbapenem-resistant strain carries no carbapenemase-encoding genes. Genetic reconstruction of the resistance phenotype confirmed that two distinct genetic loci are necessary in order for the strain to acquire carbapenem resistance. Experimental evolution of the carbapenem-resistant strains in growth conditions without the antibiotic revealed that both loci confer a significant cost and are readily lost by de novo mutations resulting in the rapid evolution of a carbapenem-sensitive phenotype. To explain how carbapenem resistance evolves via multiple, low-fitness single-locus intermediates, we hypothesised that one of these loci had previously conferred adaptation to another antibiotic. Fitness assays in a range of drug concentrations show how selection in the antibiotic ceftazidime can select for one gene ($bla_{DHA-1}$) potentiating the evolution of carbapenem resistance by a single mutation in a second gene (*ompK36*). These results show how a patient's treatment history might shape the evolution of antibiotic resistance and could explain the genetic basis of carbapenem-resistance found in many enteric-pathogens.

## Editor's evaluation

In this study, the authors examine the mechanisms of resistance to carbapenem in *Klebsiella* quasipneumoniae, through a non–carbapenemase mechanism. The evidence – which includes a combination of experimental and computational approaches – is compelling. It offers a set of

findings that make a very important contribution to our understanding of not only how antimicrobial resistance evolves, but how to integrate methods and data of different kinds towards understanding a complex evolutionary phenomenon.

## Introduction

Understanding the genetic and evolutionary provenance of antimicrobial resistance (AMR) will be important for developing strategies that slow or prevent the evolution of untreatable pathogens. In the case of bacterial pathogens, there is mounting evidence that the source of antibiotic resistance can be the patient's own microbiome (*Stracy et al., 2022*) and that treatment history, as well as pathogen genotype, should be taken into account when designing a treatment plan. Carbapenems are a class of β-lactam antibiotics typically reserved for high-risk, multidrug-resistant infections (*Doi, 2019*; *Meletis, 2016*). Surveillance studies show that carbapenem-resistant Enterobacteriaceae (CRE), especially various species of the pathogen *Klebsiella,* have become a widespread problem in clinical settings around the globe (*Hansen, 2021*; *Hu et al., 2016*; *Logan and Weinstein, 2017*).

There are two general mechanisms of carbapenem resistance. The first and readily diagnosable mechanism is the acquisition of a single gene encoding a carbapenemase enzyme that directly inactivates the antibiotic. The second mechanism requires multiple genetic loci, and bacterial strains in this category often have an over expressed β-lactamase and/or an overexpressed efflux pump and/or an inactivated outer membrane protein (*Meletis, 2016*; *Codjoe and Donkor, 2017*; *Elshamy and Aboshanab, 2020*), and potentially as yet unidentified loci as well. Of these two mechanisms, most carbapenem-resistant *Klebsiella pneumoniae* that have been identified so far are caused by carbapenemases. Like other β-lactamases, carbapenemases can be recognised by sequence analysis (*Wang et al., 2021*), and there are two structurally distinct classes of carbapenemases: the first is the metallo-β-lactamases a family of enzymes that contains a metal ion (usually zinc) coordinated in the active site, a classic example of which is NDM-1 (*Khan et al., 2017*). The second carbapenemase family does not coordinate a metal ion but instead relies on an active site serine to hydrolyse carbapenem drugs (*Codjoe and Donkor, 2017*; *Nordmann and Poirel, 2019*). This second class of carbapenemase includes the IMI, OXA-48, and GES enzymes in addition to the archetypal *K. pneumoniae* carbapenemases: the KPC enzymes (*Nordmann and Poirel, 2019*; *Rapp and Urban, 2012*; *Yigit et al., 2001*) that include the prevalent KPC-2. The KPC-2 carbapenemase has become widespread, found in many clinical investigations of CRE, and the gene encoding this enzyme - $bla_{KPC-2}$ - is transferred readily by horizontal gene transfer via plasmids (*Bowers et al., 2015*; *Hardiman et al., 2016*; *Kitchel et al., 2009*; *Petrella et al., 2008*; *Wyres et al., 2019*). Since this is a monogenic phenotype, carbapenem resistance caused by the presence of a carbapenemase is readily diagnosed by whole genome sequencing or even simple PCR-based genome tests.

It has recently become apparent that numerous CRE infections do not depend on the expression of carbapenemases, and there is mounting evidence that these non-carbapenemase CRE are widespread (*Bouganim et al., 2020*). Based on the limited cases identified for non-carbapenemase CRE so far, it has been suggested that they emerge because of reduced outer membrane permeability and/or increased drug efflux (*Logan and Weinstein, 2017*; *Goodman et al., 2016*; *Tamma et al., 2017a*). Other studies have suggested that, at least in some genetic backgrounds, an extended-spectrum β-lactamase (ESBL) could provide sufficient activity against carbapenems as to generate a CRE phenotype (*Codjoe and Donkor, 2017*; *Halat et al., 2016*). Despite this emerging knowledge, few studies have directly demonstrated the cause of non-carbapenemase CRE, and the evolutionary forces that shape the evolution of this trait have not been addressed. Understanding the details of these evolutionary forces could provide a means to re-sensitise populations of bacteria to carbapenems.

A patient died as a result of septicaemia caused by a *Klebsiella* isolate, FK688, where the infection did not respond to treatment with carbapenems (*Bi et al., 2017*). In this study, we present the complete genome sequence of *Klebsiella* FK688, using a compilation of short- and long-read sequence data (*Figure 1A*). We identify, and experimentally confirm the genetic basis of the non-carbapenemase CRE phenotype in FK688: carriage of a mega-plasmid (pNAR1) and an inactivating mutation in the chromosomal gene *ompK36*. We show that non-carbapenemase CRE strains are unfit and readily evolve to be drug-sensitive in the absence of carbapenem antibiotics. The evolution of

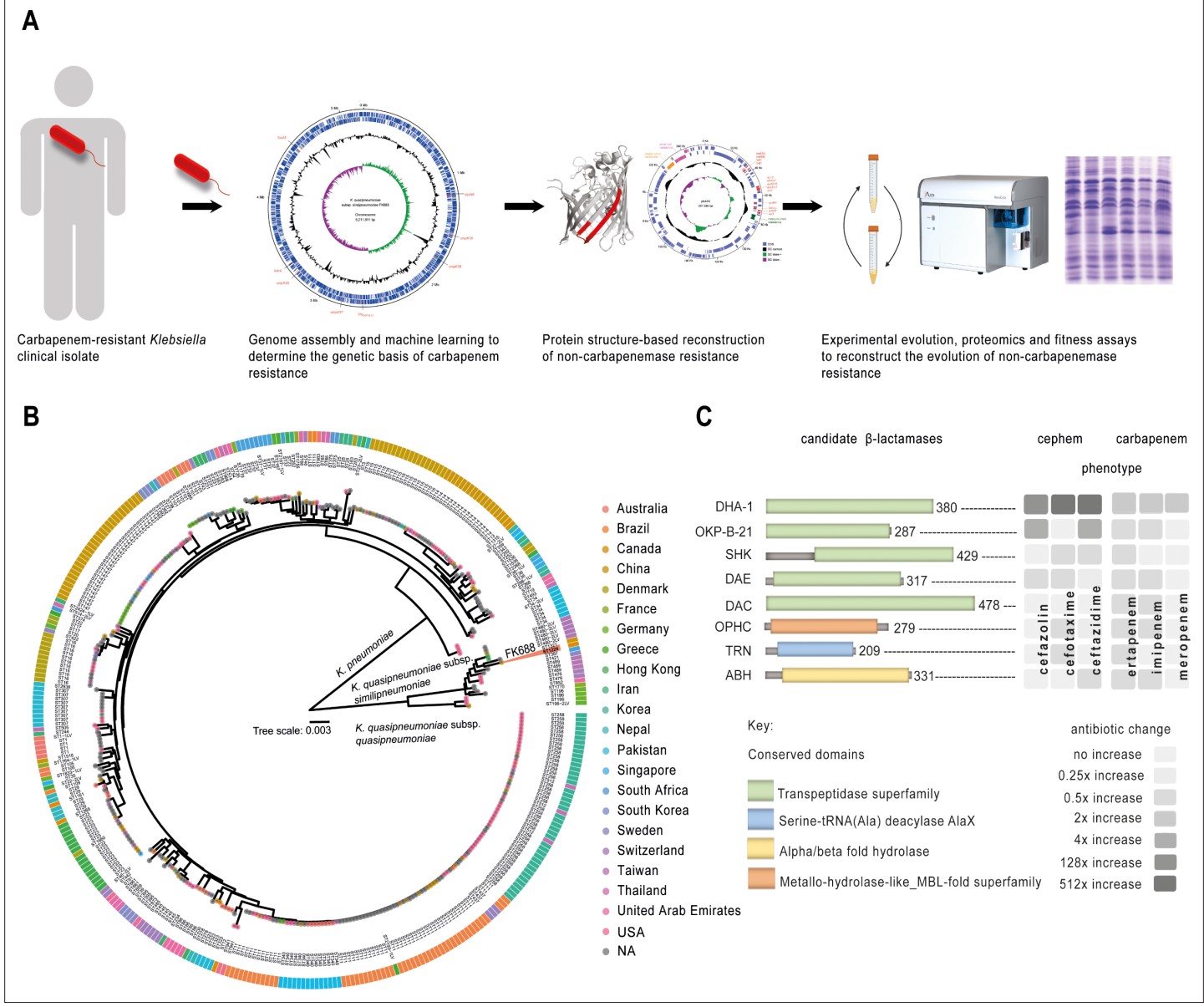

**Figure 1.** Experiment overview. (**A**) Carbapenem-resistant *Klebsiella* spp. were isolated from a patient, and the genome was sequenced and assembled. The genetic cause of resistance was confirmed by re-engineering the carbapenem resistance, partly based on structure guided restoration of a partially truncated membrane protein. The evolutionary drivers of resistance and sensitivity were determined using experimental evolution and extensive phenotypic and genotypic measures of evolutionary change. (**B**) Maximum likelihood phylogenetic tree of 377 publicly available *Klebsiella* genomes shows *K. pneumoniae* and *K. quasipneumoniae* as distinct species. The inner ring colours refer to the country of isolation according to the key, and further data is described in *Figure 1—figure supplement 2* and *Figure 1—source data 1*. (**C**) Eight candidate carbapenemases were identified in the FK688 genome sequence and overexpressed in an *E. coli* model of resistance. Only two enzymes (DHA1 and OKP-B-21) conferred resistance to the cephem antibiotics tested, and none of the enzymes conferred resistance to the carbapenem antibiotics (*Figure 1—source data 6* and *Figure 1—figure supplement 3*).

The online version of this article includes the following source data and figure supplement(s) for figure 1:

**Source data 1.** Strain information for *Figure 1B*.

**Source data 2.** Growth rate analysis of *Escherichia coli* BW25113 strains expressing the indicated open-reading frames cloned into plasmid pJP-CmR.

**Source data 3.** Antibiotic resistance genes identified in pNAR1.

**Source data 4.** Transmembrane transporter systems identified in pNAR1.

**Source data 5.** β-lactamase prediction and classification using DeepBL.

**Source data 6.** Minimum inhibitory concentration (MIC) analysis of *Escherichia coli* BW25113 expressing DeepBL candidates and MIC analysis of FK688

*Figure 1 continued on next page*

*Figure 1 continued*

pNAR1ΔblaDHA-1 strains expressing DHA-1.

**Figure supplement 1.** Physical map of the *K. quasipneumoniae* subsp. *similipneumoniae* FK688 chromosome.

**Figure supplement 2.** The phylogeny of *Klebsiella*.

**Figure supplement 3.** Domain architecture of DeepBL candidates.

these low-fitness CRE strains may be contingent on their recent exposure to antibiotics that select for other β-lactamases which may explain the evolution of non-carbepenemase CRE.

## Results

### *Klebsiella* FK688 does not encode a carbapenemase

First, to determine the genetic basis of carbapenem resistance in FK688 (*Table 1*), we used a long- and short-read sequencing approach to generate a complete assembly of the FK688 genome (*Figure 1A*, *Figure 1—figure supplement 1*). The genome assembly revealed a circular chromosome (5,211,811 bp) and a novel circular megaplasmid (pNAR; 1,257,585 bp). This plasmid carried many antibiotic resistance genes corresponding to the known resistance profile (*Bi et al., 2017*), as well as genes encoding efflux pumps and other transporters (*Figure 1—source data 3*, *Figure 1-source data 4*). Phylogenetic analysis placed FK688 within *K. quasipneumoniae* subsp. *similipneumoniae* (*Figure 1B*, *Figure 1—figure supplement 2*, *Figure 1—source data 1*). Readily identifiable determinants of AMR on the FK688 chromosome are a $bla_{OKP-B-21}$ gene encoding a β-lactamase that confers resistance to penicillins and cephalosporins such as cefazolin, as well as determinants for quinolone (*oqxA*, *oqxB*; *Brisse et al., 2014*; *Mathers et al., 2019*; *Rodrigues et al., 2019*) and fosfomycin (*fosA5*) resistance (*Figure 1—figure supplement 1*).

Our genome sequence analysis did not reveal a gene encoding KPC-2, the carbapenemase found in many *K. quasipneumoniae* isolates (*Mathers et al., 2019*). To identify any cryptic carbapenemases that may have escaped annotation, we made use of the machine learning predictor DeepBL (*Wang et al., 2021*). DeepBL identifies genes encoding β-lactamases of all types, including carbapenemases, and generated eight high-confidence predictions. The two highest predictions represent known

**Table 1.** Antimicrobial susceptibility profiling of *K. quasipneumoniae* FK688.

| Antimicrobial Class | Antimicrobial Drug | MIC (µg/mL)* | | |
|---|---|---|---|---|
| | | FK688 | *E. coli* (ATCC 25922) | Breakpoints[†] |
| Penicillins | Ampicillin | >2048 | *8* | ≥32 |
| Cephems | Cefazolin | >2048 | *2* | ≥8 |
| | Cefotaxime | 1024 | *0.125* | ≥4 |
| | Ceftazidime | >2048 | *0.5* | ≥16 |
| Carbapenems | Ertapenem | 64 | *0.016* | ≥2 |
| | Imipenem | 8 | *0.25* | ≥4 |
| | Meropenem | 4 | *0.03* | ≥4 |
| Lipopeptides | Polymyxin B | 4 | *2* | ≥4 |
| Aminoglycosides | Gentamicin | *1* | *2* | ≥16 |
| | Tobramycin | *1* | *2* | ≥16 |
| | Kanamycin | *2* | *4* | ≥64 |
| Tetracyclines | Tetracycline | 128 | *1* | ≥16 |
| Fluoroquinolones | Ciprofloxacin | *1* | *0.016* | ≥1 |

*Drug-sensitive, *italics*; drug-resistant, **bold**-text.

[†]Resistant clinical breakpoint for *Enterobacterales* given by **CLSI, 2022**.

β-lactamases $bla_{OKP-B-21}$ and $bla_{DHA-1}$. The next closest predictions were for proteins annotated as "serine hydrolase" (SHK), "MBL fold-metallo hydrolase" (OPHC), and "D-alanyl-D-alanine endopeptidase" (DAE) with conserved domain architecture retrieval tool predictions, suggesting that our search was sufficiently broad to identify plausible candidates (*Figure 1C* and *Figure 1—source data 5*). To test for carbapenemase activity, we expressed each of the eight genes in *Escherichia coli* and measured growth (*Figure 1—figure supplement 3*, *Figure 1—source data 2*). Minimum inhibitory concentration (MIC) assays of *E. coli* expressing the DeepBL candidates showed that only OKP-B-21 and DHA-1 have β-lactamase activity that includes significant resistance to ceftazidime, a third-generation cephalosporin (*Figure 1C*, *Figure 1—source data 6*). However, neither OKP-B-21 nor DHA-1 - nor any of the other proteins tested - provided resistance to carbapenems. Taken together with the genome sequence analysis, these data support that the observed carbapenem-resistant phenotype for FK688 is not caused by a carbapenemase.

## The major porins permit carbapenem sensitivity in *K. quasipneumoniae*

Porins are β-barrel proteins that transport nutrients across the outer membrane of Gram-negative bacteria but can also admit antibiotics into the bacterial cell (*Martínez-Martínez et al., 1996*; *Nicolas-Chanoine et al., 2018*; *Rocker et al., 2020*). *K. pneumoniae* has four genes encoding the major porins OmpK35, OmpK36, OmpK37, and OmpK38 (*Rocker et al., 2020*), and the position and synteny of each gene in FK688 are conserved across *K. quasipneumoniae* (*Figure 2—figure supplement 1*, *Figure 2—figure supplement 2*). Inspection of the predicted protein sequences encoded by the four genes in FK688 revealed a 1.3 kb transposase gene (IS*4* family) insertion within the 5' end of the *ompK35* gene (*Figure 2—figure supplement 1*) and a 48 bp in-frame deletion in *ompK36* in FK688 (*Figure 2—figure supplement 1*).

The structure of OmpK36 (PDB 5O79) is known (*Acosta-Gutiérrez et al., 2018*), and the identified deletion of 16 amino acids at the 3' end of the *ompK36* gene in FK688 encompasses large portions of the β14 and β15 strands of the β-barrel structure (*Figure 2A* and *Figure 2B*), explaining why the protein is not assembled into the outer membrane of FK688. To confirm that this mutation was contributing to carbapenem resistance, we used the ompK36 gene from *K. quasipneumoniae subsp.* similipneumoniae ATCC 700603 (*Elliott et al., 2016*) to carry out structure-informed repair of the OmpK36 gene in *K. quasipneumoniae* FK688 (*Figure 2C*). The resultant strain (*ompK36⁺* pNAR1) was subjected to immunoblotting with antisera, confirming the restoration of OmpK36 expression in the *ompK36⁺* pNAR1 strain (*Figure 2D*, *Figure 2—source data 1* and *Figure 2—source data 2*). Finally, measurements of carbapenem MIC determined that the repaired gene encoding the porin OmpK36 restored carbapenem sensitivity (*Table 2*).

## DHA-1 and Δ*ompK36* are required for carbapenem resistance and impose non-additive fitness costs in growth media without antibiotics

To determine the evolutionary stability of the pNAR1 plasmid, we passaged 10 replicate mutation accumulation lines of *K. quasipneumoniae* FK688 in growth media without β-lactam antibiotic selection (*Figure 3A*). After 11 passages, two replicates had completely lost resistance to the β-lactam antibiotic ceftazidime. The first lineage lost a 17 kb region of pNAR1 that included the $bla_{DHA-1}$ and *qnrB4* antibiotic resistance genes flanked by the gene mobility elements *tnpA-sul1* (pNAR1Δ$bla_{DHA-1}$) while the second lineage lost the entire pNAR1 plasmid (pNAR1⁻; *Figure 3B*, *Figure 3C*). We assayed both the pNAR1⁻ and pNAR1Δ$bla_{DHA-1}$ strains for growth (*Figure 3—figure supplement 2*, *Figure 3—source data 1*) and antibiotic sensitivity (*Table 2*) and confirmed that loss of pNAR1, and specifically the *tnpA-sul1* region of pNAR1, caused a loss of resistance to ceftazidime and carbapenem.

Whole genome sequencing established that the pNAR1⁻ and pNAR1Δ$bla_{DHA-1}$ strains had not sustained any other mutations, which confirmed the genotypes of strains Δ*ompK36* and *ompK36⁺* with and without the plasmid-encoded $bla_{DHA-1}$ gene. We therefore engineered repaired versions *ompK36⁺* for each strain and tested each mutant for growth and carbapenem resistance (*Table 2*). The results demonstrate that the acquisition of carbapenem-resistance in FK688 required the combination of (i) the absence of porins and (ii) the β-lactamase DHA-1 (*Figure 1—source data 6*).

While the expression of DHA-1 and the absence of a functional porin provides a selective advantage to *Klebsiella* in high concentrations of carbapenem antibiotics, we sought to understand whether these genotypes impose a fitness cost in environments without carbapenems. Competitive fitness

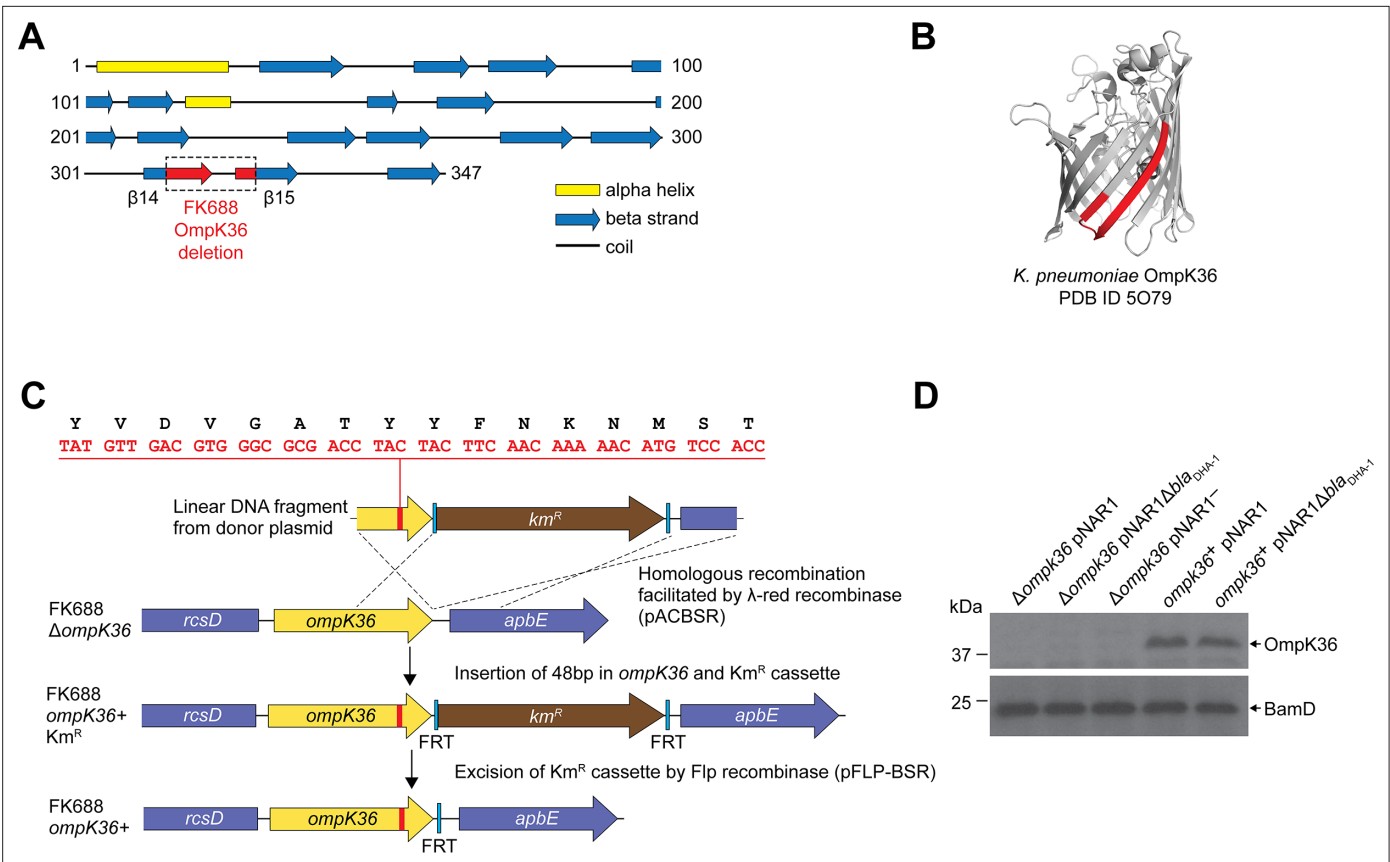

**Figure 2.** Reconstruction of mutant OmpK36 protein based structural characteristics of OmpK36 in *K. quasipneumoniae* subsp. *similipneumoniae*. (**A**) PSIPRED (**Buchan and Jones, 2019**) secondary structure prediction of the OmpK36, using the protein sequence encoded in the *K. quasipneumoniae* subsp. *similipneumoniae* genome. The location of the 16 amino acid deleted region in FK688 OmpK36 is highlighted in red on the β14-β15 strands of the corresponding structural model. (**B**) Tertiary structure of the β-barrel OmpK36 monomer (PDB ID 5O79 **Acosta-Gutiérrez et al., 2018**). Coloured red are the β14-β15 strands, the same region designated with red colour in panel (**A**). (**C**) Schematic depicting the engineering to restore a functional version of *ompK36* in FK688 by the insertion of 48 nucleotides in *ompK36*, as shown in red. FRT (flippase recognition target) sites permitted excision of the Km$^R$ (kanamycin resistance) cassette using Flp recombinase. Following Km$^R$ excision, a single FRT site and scar region remain in between the *ompK36* and *apbE* genes. The amino acid identity between the OmpK36 from ATCC 700603 and FK688 is 95% (**Figure 2—figure supplement 2**), and the ATCC 700603 sequence (**Elliott et al., 2016**) was used to repair the *ompK36* locus of FK688, as described in the Materials and methods section. (**D**) Total membrane extracts were prepared from the indicated strains, the proteins in the samples analysed by sodium dodecyl sulfate-polyacrylamide gel (SDS-PAGE) and immunoblotting using an antibody probe that recognises OmpK36 (**Rocker et al., 2020**). The outer-membrane protein BamD was used as a sample loading control for the analysis.

The online version of this article includes the following source data and figure supplement(s) for figure 2:

**Source data 1.** Original gel image for *Figure 2*.

**Source data 2.** Alternative gel image for *Figure 2*.

**Figure supplement 1.** Gene alignment of four major porins.

**Figure supplement 2.** Comparative sequence analysis of *ompK35* and *ompK36* genes of *Klebsiella* spp. and reference genomes.

assays were carried out to determine the fitness effects of the *bla*$_{DHA-1}$ and *ompK36* alleles alone and in combination (Materials and methods, *Figure 4A*). We found that the *bla*$_{DHA-1}$ and *ΔompK36* alleles conferred a substantial fitness cost in growth media without antibiotic as compared to the other strains tested (*Figure 4B*, *Figure 4—source data 1*). To refine our understanding in terms of carbapenem-resistance, we measured the fitness effects of the *bla*$_{DHA-1}$ and *ompK36* alleles across a range of imipenem concentrations. These assays showed that the fitness defects seen in the absence of imipenem (*Figure 4B*) are gradually reversed in the strains grown in the presence of increasing concentrations imipenem (*Figure 4C*, *Figure 4—source data 1*). At the breakpoint value of 0.125 µg/mL, three genotypes are already at a selective disadvantage: *ompK36*$^+$pNAR1Δ*bla*$_{DHA-1}$, *ompK36*$^+$pNAR1,

**Table 2.** Antimicrobial susceptibility profiling of FK688-derived strains.

| Antimicrobial Class | Antimicrobial Drug | MIC (µg/mL)* | | | | |
| | | ΔompK36 | | | ompK36⁺ | |
| | | pNAR1 | pNAR1 Δ$bla_{DHA-1}$ | pNAR1⁻ | pNAR1 | pNAR1 Δ$bla_{DHA-1}$ |
|---|---|---|---|---|---|---|
| Penicillins | Ampicillin | **>2048** | **256** | **128** | **>2048** | **128** |
| Cephems | Cefazolin | **>2048** | **32** | **32** | **>2048** | *4* |
| | Cefotaxime | **1024** | *0.5* | *0.5* | **32** | *0.125* |
| | Ceftazidime | **>2048** | *1* | *0.5* | **512** | *0.25* |
| Carbapenems | Ertapenem | **64** | *0.5* | *0.5* | *0.5* | *0.016* |
| | Imipenem | **8** | *0.25* | *0.25* | *1* | *0.12* |
| | Meropenem | **4** | *0.06* | *0.125* | *0.125* | *0.03* |
| Lipopeptides | Polymyxin B | **4** | **4** | **4** | **4** | **4** |
| Aminoglycosides | Kanamycin | *2* | **4** | *2* | **4** | **4** |
| Tetracyclines | Tetracycline | **128** | **256** | *2* | **128** | **128** |
| Fluoroquinolones | Ciprofloxacin | **1** | *0.06* | *0.06* | **1** | *0.125* |

*Drug-sensitive, *italics*; drug-resistant, **bold**-text.

and Δ*ompK36* pNAR1⁻. Above the breakpoint value at 0.25 µg/mL, all four genotypes are at a selective disadvantage relative to the parental FK688. Thus, a clinically relevant appearance of the CRE phenotype requires a combination of the presence of DHA-1 and the lack of a functional porin.

## High fitness and carbapenemase sensitivity rapidly evolve in experimental populations of FK688

Mutations that inactivate major porins restrict the permeability of the outer membrane. To address the fitness cost of major porin loss over time, we passaged 20 Δ*ompK36* pNAR1 (lineage A) and 20 *ompK36*⁺ pNAR1 (lineage B) replicate populations across 200 generations of evolution in media without antibiotics (*Figure 5A*). The ancestral FK688 strains (Δ*ompK36* pNAR1 and *ompK36*⁺ pNAR1) have an opaque colony morphology. This is consistent with a phenotype of capsular polysaccharide production: non-fimbriated *Klebsiella* strains are mucoid due to capsule secretion and look opaque, while translucent colonies are fimbriated and non-mucoid (*Matatov et al., 1999*; *Schembri et al., 2005*; *Wilksch et al., 2011*).

The Δ*ompK36* pNAR1 (lineage A) populations evolved similar competitive fitness to the *ompK36*⁺ pNAR1 (lineage B) evolved populations, recovering the fitness cost of the Δ*ompK36* mutation (*Figure 5C*, *Figure 5—source data 1*). To understand the molecular basis of these phenotypes, whole genome sequence data was analysed for four evolved strains: A2(o), A2(t), B3(o), and B3(t). Each, independently, sustained a deletion in pNAR1 that removed *bla*$_{DHA-1}$. This explains the increased carbapenem sensitivity of the Δ*ompK36* pNAR1 and *ompK36*⁺ pNAR1 lineages as they had now lost one of the genes contributing to provide the phenotype. After 200 generations of evolution, samples were plated on agar and were found to display a mixture of opaque and translucent colony morphotypes from each strain (*Figure 5B* and *Figure 5D*). This feature has been seen before in *Klebsiella* spp. and depends on the expression of type 1 fimbriae (*Matatov et al., 1999*; *Klemm, 1986*; *Struve et al., 2008*).

The replicate population FK688 Δ*ompK36* pNAR1(A2) population evolved to become a mixture of opaque (o) and translucent (t) colony-forming types, and we chose one of each colony morphotype A2(o) and A2(t) for a more detailed analysis, while for the replicate population *ompK36*⁺ pNAR1 (B3) was almost entirely (594 of 595 colonies observed) made up of translucent colony-forming cells, we selected one of the translucent colonies B3(t) and recovered the only opaque colony B3(o) for further analysis (*Figure 5D*, *Figure 5—source data 2*). In the single B3(o) isolate recovered, a transposase gene (IS4 family) insertion had disrupted the downstream region of the *fimE* gene (*Figure 5—figure*

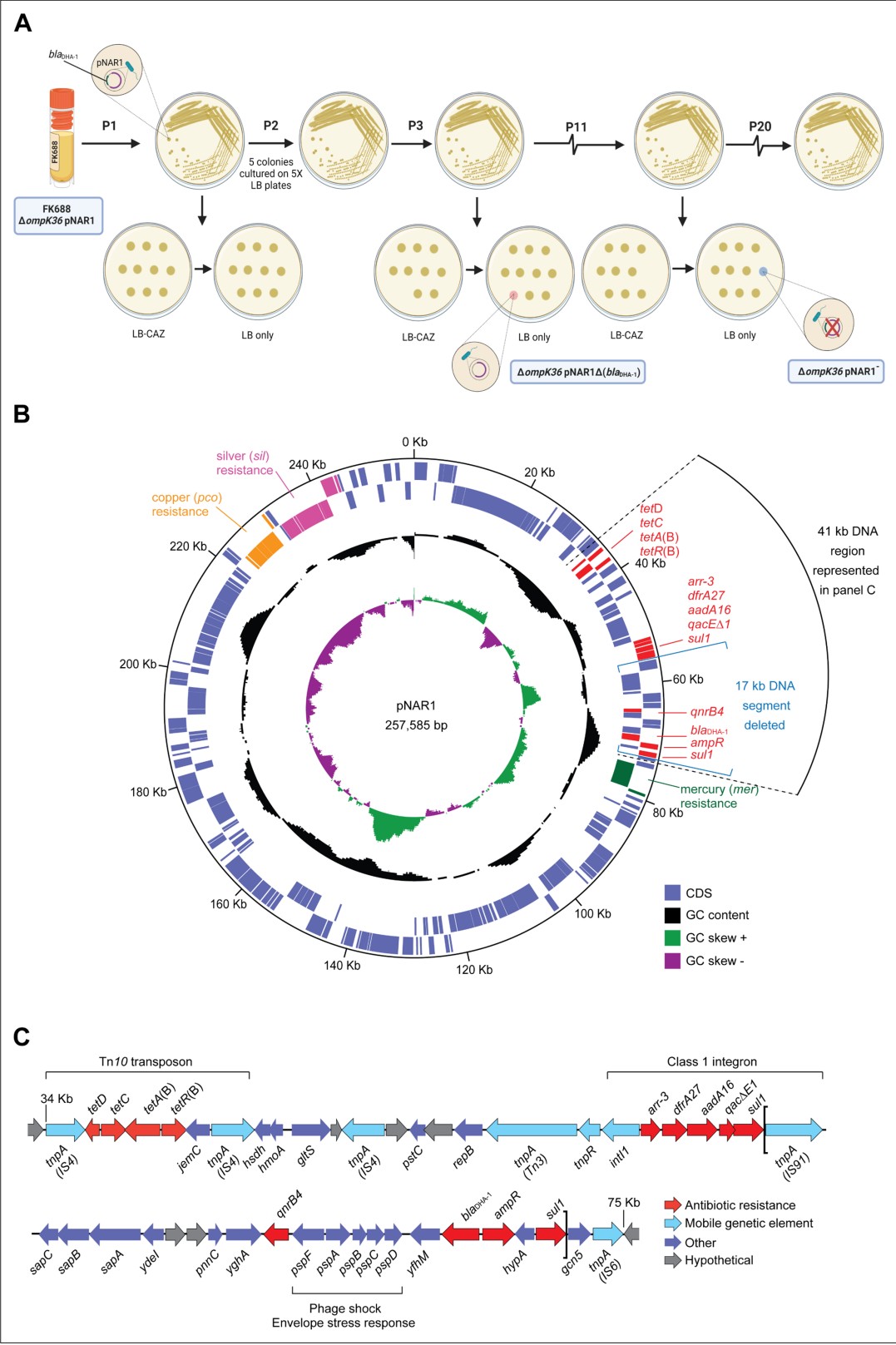

**Figure 3.** Evolution and physical map of plasmid pNAR1. (**A**) Schematic representation of the in vitro evolution experiment. After passage #3 (P3), a ceftazidime-susceptible (CAZ$^S$) mutant evolved, lacking a 17 kb region of pNAR1 that included $bla_{DHA-1}$ (referred to as Δ$ompK36$ pNAR1Δ($bla_{DHA-1}$)). After 11 passages (P11) a CAZ$^S$ colony missing the entire plasmid (referred to as Δ$ompK36$ pNAR1$^-$) evolved. In total, 20 passages were performed, and

*Figure 3 continued on next page*

*Figure 3 continued*

another five CAZ$^S$ colonies were identified, each missing the 17 kb region of pNAR1 that includes $bla_{DHA-1}$. (**B**) The position of genes encoding antibiotic resistance determinants (red), and efflux pumps annotated as being for mercury resistance (green), copper resistance (orange), and silver resistance (pink) are indicated. In addition to $bla_{DHA-1}$, pNAR1 carries genes encoding AmpR (ID00077) a transcriptional regulator known to regulate expression of $bla_{DHA-1}$ (*Realegeno et al., 2021*). Also, other drug resistance genes including those responsible for resistance to tetracycline (*tetA*[B]), rifamycin (*arr-3*), trimethoprim (*dfrA27*), streptomycin (*aadA16*), macrolides (*qacΔE1*), sulfonamides (*sul1*), and quinolones and fluoroquinolones (*qnrB4*; *Figure 3—figure supplement 1*, *Figure 1— source data 3*). The location of predicted coding sequences in the forward (outer most) and reverse DNA strands is designated by purple boxes in the outer concentric circles. The middle circle (black) graphs the % GC content, and the inner circle indicates the positive (green) and negative (purple) GC skew ([G−C]/[G+C]). The map was generated with DNAPlotter (*Carver et al., 2009*). The black arc designates a 41 kb segment of DNA expanded in panel C. (**C**) Linear map of the 41 kb segment of pNAR1 showing the genetic arrangement of antimicrobial resistance genes (red), mobile genetic elements (blue), annotated coding sequences (purple), and hypothetical genes of unknown function (grey). Assigned IS families are shown underneath each transposase gene (*tnpA*). The loci within the two brackets represent the 17 kb DNA segment (*tnpA-sul1*) deleted from pNAR1Δ$bla_{DHA-1}$.

The online version of this article includes the following source data and figure supplement(s) for figure 3:

**Source data 1.** Growth rate analysis of *K. quasipneumoniae* strains.

**Figure supplement 1.** Detailed physical map of the FK688 plasmid pNAR1.

**Figure supplement 2.** Growth rate analysis of *K. quasipneumoniae* strains.

---

*supplement 1*, *Figure 5—source data 3*). The *fimE* gene was not disrupted in the A2 population that was capable of phase-switching (*Figure 5—figure supplement 2*). Consistent with these observations, an assay measuring glucuronic acid that reflects the presence of capsule showed that the A2(o) and B3(o) had more capsular polysaccharide than A2(t) and B3(t) (*Figure 5E*, *Figure 5—source data 4*).

MIC analysis of the lineage A evolved strains showed that the Δ*ompK36* pNAR1 populations had evolved increased sensitivity to imipenem, 64-fold for the A2(o) and A2(t) strains, bringing them to the same MIC value as the B3(t) strain (*Table 3*). Protein analysis of A2(o) and A2(t) by sodium dodecyl sulfate-polyacrylamide gel (SDS-PAGE; *Figure 5F*, *Figure 5—source data 5*) and mass spectrometry showed no changes that were consistent in both populations to explain the increased sensitivity to imipenem. Our reconstruction experiments (*Table 2*) suggest that the primary determinant of this reversion of the AMR phenotype is the observed loss of the $bla_{DHA-1}$ gene from the megaplasmid pNAR1.

## Non-carbapenemase carbapenem resistance evolves via ceftazidime resistance

The results so far confirm that two genetic variants are required for the evolution of non-carbapenemase resistance to imipenem. The evolution of this trait is puzzling because both the Δ*ompK36* strain and the $bla_{DHA-1}$ positive strain each have a low fitness in growth media without antibiotic, as well as in growth media with imipenem. One explanation for the evolution of carbapenem resistance in a strain of *Klebsiella* with an intact major porin is the simultaneous acquisition of multiple genetic variations after the population was exposed to imipenem. However, given that the two genes are not linked, the simultaneous acquisition of a new gene ($bla_{DHA-1}$) and a spontaneous genetic variant (inactivating mutation in one or more major porins such as *ompK36*) is highly unlikely.

An alternative explanation is that the bacterial population may have first been subjected to environmental conditions that selected for the fixation of one of the alleles. For antibiotic resistance phenotypes, this is a realistic scenario. Given the difficulty with diagnosing a non-carbapenemase CRE infection, patients might first be treated with other β-lactams. To address this possibility, we tested whether an antibiotic other than imipenem could have selected for the presence of one or both alleles, thus potentiating the evolution of imipenem resistance with a single mutational step.

The fitness of each combination of the $bla_{DHA-1}$ and *ompK36* alleles was tested in a range of ceftazidime concentrations (*Figure 6A*, *Figure 6—source data 1*). This showed that $bla_{DHA-1}$ was selectively favoured, even at concentrations of ceftazidime approximately 100× below the clinical breakpoint of 16 μg/mL, despite the fact that carriage of the $bla_{DHA-1}$ gene bears a substantial fitness cost

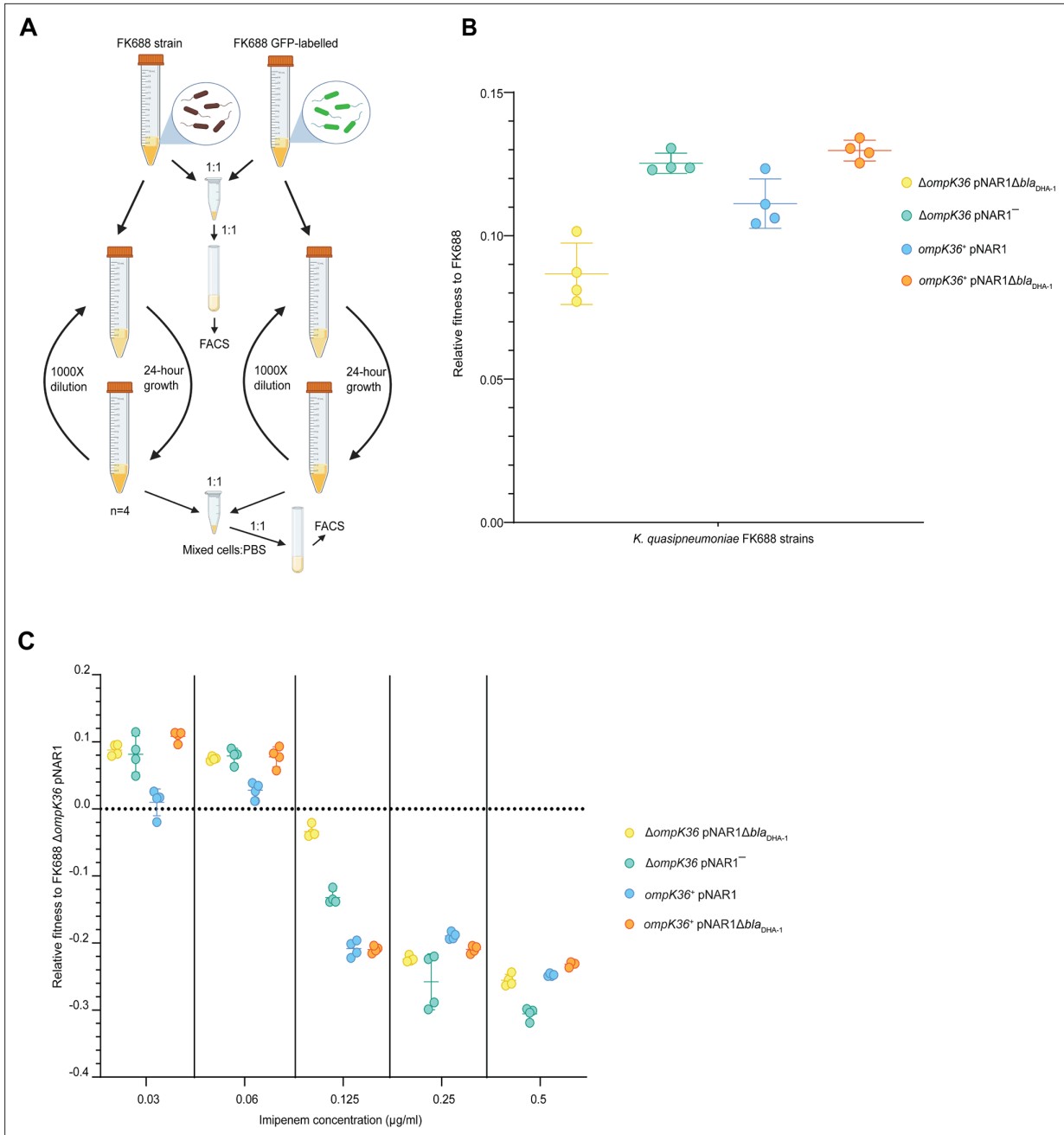

**Figure 4.** Competitive fitness assay of FK688 strain variants against GFP-labelled FK688. (**A**) Schematic of the competitive fitness assay experiment (Materials and methods). FACS: fluorescence-activated cell sorting. (**B**) The relative fitness of the engineered mutant strains relative to the carbapenem resistant FK688 strain, measured in Lysogeny Broth (LB) growth media without antibiotics. Mutant strains either have the OmpK36 outer membrane transporter restored (*ompK36*⁺), the DHA-1 β-lactamase deleted (pNAR1⁻ or pNAR1Δ*bla*_DHA-1_), or both (orange circles). The y-axis shows the selection coefficient (S) per generation compared to the carbapenem resistant ancestor FK688 which has its fitness set at 0. The legend indicates the genotypes for each strain (note FK688 is genotype Δ*ompK36* pNAR1). Error bars represent mean ± SD (n=4). (**C**) Relative fitness of FK688 mutant strains compared to parental FK688, measured in LB media supplemented increasing concentrations of the carbapenem antibiotic imipenem. The legend shows the genotype for each *Klebsiella* strain. Error bars represent mean ± SD (n=4).

The online version of this article includes the following source data for figure 4:

**Source data 1.** Relative fitness values for each data point in *Figure 4B and C*.

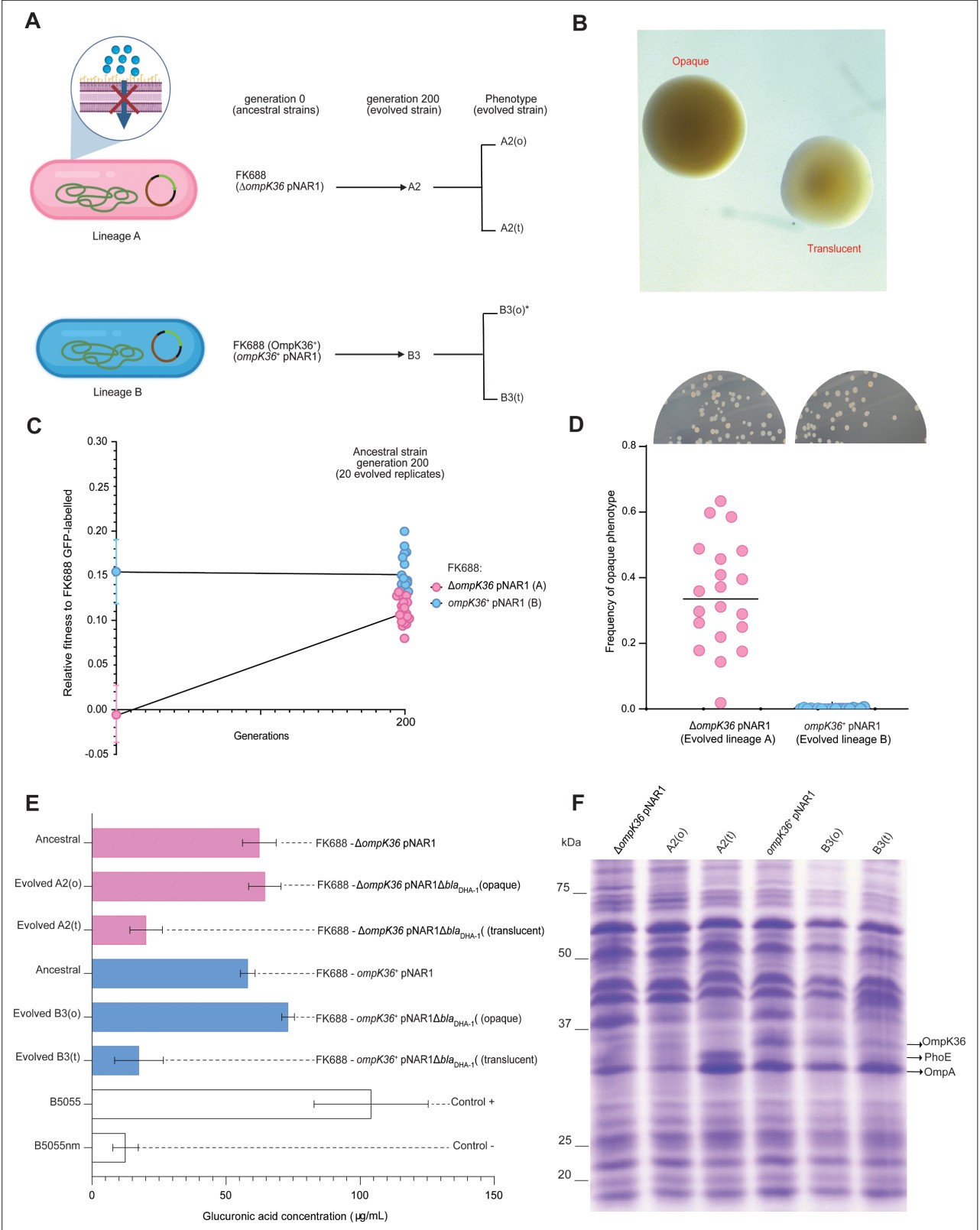

**Figure 5.** Genotypic and phenotypic evolution of FK688 Δ*ompK36* and *ompK36*⁺strains. (**A**) Schematic of the evolution experiment of Lineage A (FK688:Δ*ompK36* pNAR1) and Lineage B (FK688:*ompK36*⁺pNAR1). 20 replicate populations (A1, A2, A3,…A20 and B1, B2, B3,…B20) for each lineage were serially passaged (1000-fold dilutions at each passage) for 200 generations. Of the evolved strains, population A2 and population B3 were characterised as described in the text. While population A2 showed a mixture of both opaque (o) and translucent (t) colony morphotypes, the

*Figure 5 continued on next page*

*Figure 5 continued*

asterisk (*) denotes that only a single colony of opaque (o) morphotype was observed in population B3. (**B**) Colony morphotypes seen in the evolved populations. Colonies were grown on 0.5✕ Lysogeny Broth (LB) agar for an overnight incubation at 37°C and photographed with stereo microscope using transmitted light to capture translucency. (**C**) The relative fitness assessments for the populations of FK688 Δ*ompK36* pNAR1(pink) and populations of *ompK36*+pNAR1 (blue) genotypes (left). The error bars represent mean ± SD (n=4). Relative fitness assays were also performed for 20 evolved populations after 200 generations of evolution in LB growth media without antibiotics (right). The line represents individual replicates with means connected. (**D**) Relative numbers of opaque colonies in the 20 replicate populations of FK688 Δ*ompK36* pNAR1 (lineage A) and *ompK36*+pNAR1 (lineage B) strains after 200 generations. Each dot represents an individually evolved population. The inset photographs (above the graph) are an example of frequency of opaque and translucent colonies on an agar plate in one of the 20 replicate populations. (**E**) Capsular polysaccharide was extracted from cell cultures for glucuronic acid measurement (Materials and methods; *Campos et al., 2004*). The error bars represent mean ± SD (n=3). For reference, ancestral and evolved strains were compared with the hypermucoid (i.e. heavily capsulated) clinical isolate B5055 and an isogenic mutant B5055nm Δ*wza-wzc* (non-mucoid) strain. (**F**) Total cell extracts from the indicated strains were analysed by sodium dodecyl sulfate-polyacrylamide gel (SDS-PAGE) and Coomassie staining. The migration positions of OmpK36, PhoE, and OmpA are indicated. The identities of these protein species were confirmed by mass spectrometry of the corresponding region of the gel.

The online version of this article includes the following source data and figure supplement(s) for figure 5:

**Source data 1.** Fitness assay ancestral and evolved lineages A and B strains.

**Source data 2.** Relative numbers of opaque colonies in the 20 replicate populations of FK688 ΔompK36 pNAR1 (lineage A) and ompK36+pNAR1 (lineage B) strains after 200 generations.

**Source data 3.** FK688 OmpK36+, B3(o), and B3(t) genome modification and SNP analysis.

**Source data 4.** Glucuronic acid measurement of ancestral and evolved strains.

**Source data 5.** Original gel image for *Figure 5*.

**Figure supplement 1.** Comparative sequence analysis of the *fim* gene cluster in evolved *K. quasipneumoniae* strains.

**Figure supplement 2.** Comparative sequence analysis of the *fim* gene cluster in evolved *K. quasipneumoniae* strains.

(*Figure 4B*). In addition, we found that the OmpK36 inactivating mutation reduced fitness in ceftazidime, confirming that selection on ceftazidime can select for the *bla*$_{DHA-1}$ gene, but not for the *ompK36* mutation, which is required for imipenem resistance.

These results support the hypothesis that imipenem resistance evolved in FK688 via multiple evolutionary steps. First, the population experienced an antibiotic-containing environment that selected for the *bla*$_{DHA-1}$ gene. This could have been any antibiotic that selected for the pNAR1 plasmid, such

**Table 3.** Antimicrobial susceptibility profiling FK688 Δ*ompK36* and *ompK36*+ strains and their respective evolved strains.

| | | MIC (µg/mL)* | | | | | |
|---|---|---|---|---|---|---|---|
| | | **Δ*ompK36*** | | | ***ompK36*+** | | |
| **Antimicrobial Class** | **Antimicrobial Drug** | **pNAR1** | **pNAR1 Δ*bla*$_{DHA-1}$ A2(o)** | **pNAR1 Δ*bla*$_{DHA-1}$ A2(t)** | **pNAR1** | **pNAR1 Δ*bla*$_{DHA-1}$ B3(o)** | **pNAR1 Δ*bla*$_{DHA-1}$ B3(t)** |
| Cephems | Cefazolin | >2048 | 32 | 1 | >2048 | *2* | *1* |
| | Cefotaxime | 1024 | *0.5* | *0.25* | 32 | *0.125* | *0.25* |
| | Ceftazidime | >2048 | *0.5* | *0.25* | 512 | *0.25* | *0.25* |
| Carbapenems | Ertapenem | 64 | *0.5* | *0.031* | *0.5* | *0.016* | *0.031* |
| | Imipenem | 8 | *0.125* | *0.125* | *1* | *0.25* | *0.125* |
| | Meropenem | 4 | *0.125* | *0.016* | *0.12* | *0.016* | *0.016* |
| Tetracyclines | Tetracycline | 128 | 128 | 128 | 128 | 128 | 128 |
| Fluoroquinolones | Ciprofloxacin | 1 | *0.125* | *0.25* | 1 | *0.031* | *0.063* |
| Aminoglycosides | Kanamycin | *2* | *2* | *2* | *4* | *2* | *2* |
| | Tobramycin | *0.5* | *1* | *1* | *1* | *1* | *1* |
| | Gentamicin | *0.5* | *1* | *0.5* | *0.5* | *0.5* | *0.5* |
| Lipopeptides | Polymyxin B | 4 | 4 | 4 | 4 | 8 | 4 |

*Drug-sensitive, *italics*; drug-resistant, **bold**-text.

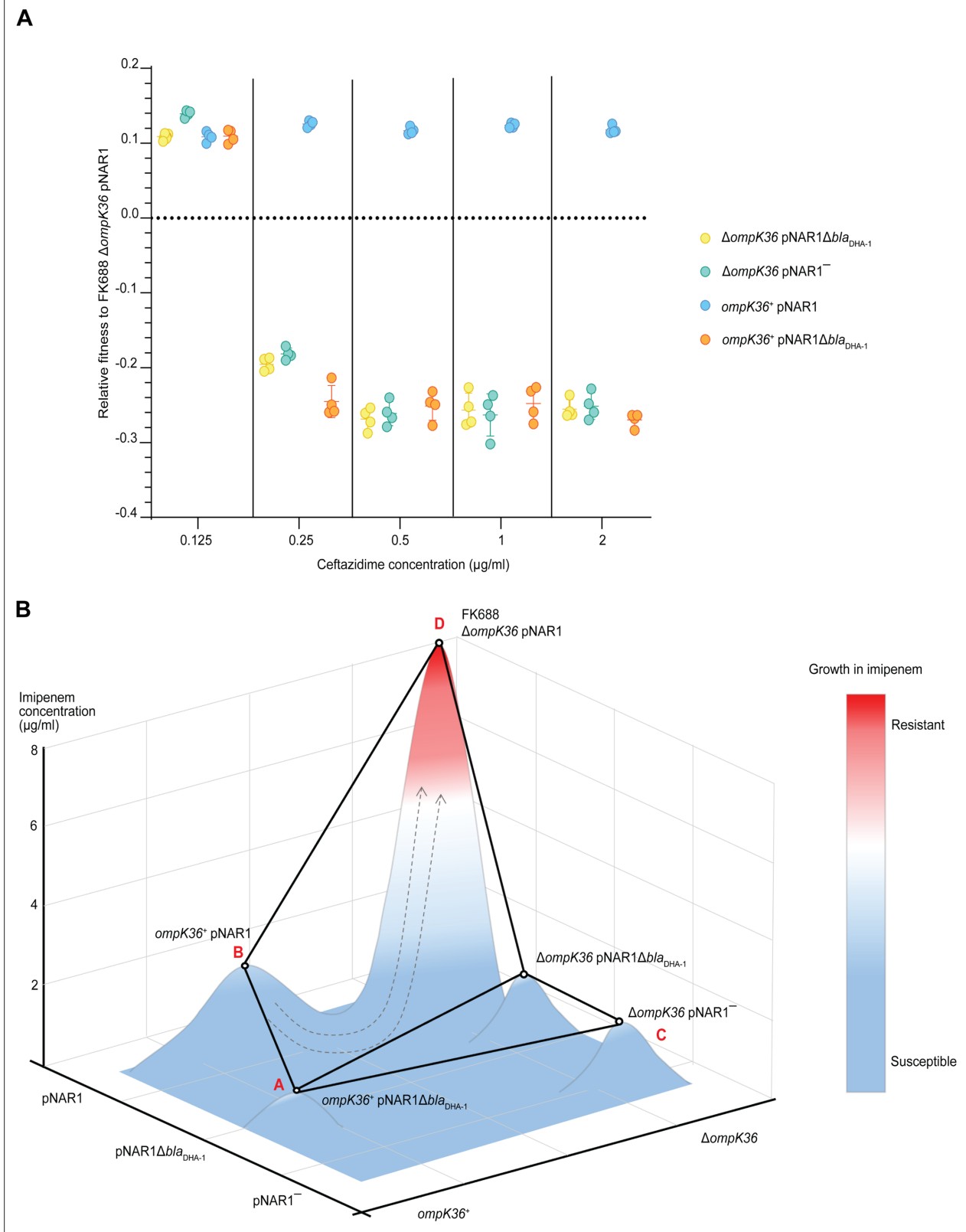

**Figure 6.** Competitive fitness assay of FK688 strain variants in the presence of ceftazidime. (**A**) The fitness of FK688 mutants measured in Lysogeny Broth (LB) media supplemented with ceftazidime across a concentration range from 0.125 to 2 µg/mL. Only strains with an intact pNAR1 plasmid, including the $bla_{DHA-1}$ gene, are able to survive high concentrations of ceftazidime. The legend has the genotypes for the *Klebsiella* strains. Error bars represent mean with ± SD (n=4). (**B**) Schematic of the imipenem resistance landscape. Each genotype is depicted as being resistant (red) or susceptible (blue) to

*Figure 6 continued*

impenem. The x and y planes depict the antimicrobial resistance (AMR) genotypes, and the z plane represents growth measured at each concentration of imipenem. Circles represent the genotype of each strain, and lines show strain connected by a single mutation. The evolution of imipenem resistance requires two genes - the $bla_{DHA-1}$ gene, and a loss of function mutation in *ompK36*: these two alleles are both found in FK688, indicated at "D". Since both single-step mutants "B" and "C" are imipenem susceptible and do not have a fitness advantage in growth media without drugs (*Figure 4B*), we propose that the population had recently been exposed to conditions that selected for the pNAR1 plasmid. Then after exposure of the population to imipenem, the Δ*ompK36* mutant was strongly selected. This suggests that the most likely evolutionary path to imipenem resistance was A → B → D.

The online version of this article includes the following source data for figure 6:

**Source data 1.** Relative fitness values for each data point in *Figure 6A*.

**Source data 2.** Data used to construct fitness landscape in *Figure 6B*.

as ceftazidime or another β-lactam antibiotic. After a short period of selection on this first treatment, most of the population would carry the $bla_{DHA-1}$ gene, increasing the chance that an *ompK36* inactivating mutation would occur in a cell that also carried the $bla_{DHA-1}$ gene. We tested the evolutionary potential for each genotype to evolve imipenem resistance by setting up cultures of the FK688 strains with either one, or none, of the two drug resistance alleles and plating $10^9$ cells of each genotype on a range of concentrations of imipenem (*Figure 6B*, *Figure 6—source data 2*). We evaluated the evolutionary path to antibiotic resistance by considering the starting point of evolution as the genotype that does not have the $bla_{DHA-1}$ gene or the loss of function mutation in *ompK36*, indicated at "A". This genotype is the most logical starting point because it has the highest fitness in growth conditions without antibiotic (*Figure 4B*). We found that strains that carried only the Δ*ompK36* allele, or neither allele, were unable to evolve imipenem resistance (*Figure 6B*; "A" and "C"). However, a *K. quasipneumoniae* strain with the pNAR1 plasmid (carrying the $bla_{DHA-1}$ gene) would be readily able to evolve resistance (*Figure 6B*; "B"). Thus, in a scenario representing previous treatment of a patient with ceftazidime, given the low fitness benefit conferred by the loss of function mutation in *ompK36* in growth media supplemented with imipenem, the path A → B → D is most likely.

## Discussion

This study presented the first physical genetic map of a *K. quasipneumoniae* subsp. *similipneumoniae* genome. In addition to species-specific AMR loci on the bacterial chromosome, this multidrug-resistant strain carries many characterised AMR traits on a megaplasmid that was named pNAR1. We have further used whole genome sequencing of *K. quasipneumoniae* populations to characterise the genetic and evolutionary mechanism with which they acquire carbapenem resistance. A major finding of the study is the ease with which carbapenem sensitivity was restored in the absence of drug selection. This bodes well for new strategies that are being devised to reverse the evolution of AMR phenotypes in *Klebsiella* spp. populations, be they in built environments, in gut microbiomes or in infection sites. A further major finding is the means by which an ill-chosen drug treatment, for example with ceftazidime, can prime a population of *K. quasipneumoniae* to rapidly evolve a CRE phenotype.

Genomics-based surveillance has shown that KPC-2 carbapenemases are widespread in *Klebsiella* spp. including *K. quasipneumoniae* (*Mathers et al., 2019*; *Octavia et al., 2019*). β-lactamases encoded by chromosomal genes are common amongst these species of *Klebsiella*: $bla_{SHV}$ is found in *K. pneumoniae*, $bla_{OKP-A}$ in *K. quasipneumoniae* subsp. *quasipneumoniae* and $bla_{OKP-B}$ in *K. quasipneumoniae* subsp. *similipneumoniae* (*Rodrigues et al., 2019*; *Long et al., 2017*; *Fevre et al., 2005*). Consistent with this, FK688 carries a chromosomally located $bla_{OKP-B}$. In addition, FK688 also carries the β-lactamase $bla_{DHA-1}$ on the megaplasmid pNAR1.

### Epistatic impacts of porins, pumps, and enzymes on carbapenem-resistance

There have been no reports on drug efflux pumps in *K. quasipneumoniae*, but in some strains of *K. pneumoniae*, drug-resistance phenotypes have been suggested to include epistatic contributions from genes encoding efflux pumps (*Nicolas-Chanoine et al., 2018*). FK688 encodes numerous ABC-type efflux systems with annotations for these efflux pumps suggestive of metal ion ligands (copper, silver, and mercury), but the ligand specificity of efflux pumps can be broader or different to that denoted by annotation (*Mata et al., 2000*; *Flach et al., 2017*). In FK688, the efflux pumps did not contribute

to carbapenem resistance. Are there any further epistatic effects relevant to imipenem sensitivity? A potential one would be that other porins had been upregulated, and the A2(t) translucent strain was observed to have increased expression of outer membrane proteins OmpA and PhoE (*Figure 5E*). However, (i) OmpA does not form a sizeable channel in the outer membrane (*Ortiz-Suarez et al., 2016*), and (ii) while the porin PhoE does form a channel (*Rocker et al., 2020*), it is not conducive to permitting imipenem entry into *Klebsiella* (*Rocker et al., 2020*). A single SNP that was fixed in the A2 population was in the *cadBA* operon that generates cadaverine (*Figure 5—source data 3*). CadC is a positive activator of the *cadBA* operon (*DelaVega and Delcour, 1995*), and this activation of *cadBA* is known to close porins in general and block β-lactam influx (*Dela Vega and Delcour, 1996*; *Iyer and Delcour, 1997*; *Samartzidou et al., 2003*), with point mutations in the CadC protein sufficient to inhibit the *cadBA* operon (*Schlundt et al., 2017*).

Instead, we found that the CRE phenotype in FK688 depends on epistasis between $bla_{DHA-1}$ and *ompK36*. Both the carriage of $bla_{DHA-1}$ and the defect in *ompK36* have measurable fitness cost to the strain. After multiple rounds of plating in the absence of drug-selection, we observed the loss of a segment *(tnpA-sul1)* of pNAR1 that carries $bla_{DHA-1}$, that is, selection against the β-lactamase. This same outcome was also observed in a controlled evolution experiment over 200 generations. These observations are explained by the measured fitness cost in pNAR1Δ$bla_{DHA-1}$ which was less than the fitness cost imparted by the full plasmid. Thus, in the absence of β-lactam, the CRE phenotype is reversed to carbapenem-sensitivity. It is not clear from current literature how widespread the non-carbapenemase CRE phenotype is, but several points are worthy of note: (i) the presence of plasmids encoding DHA-1 is geographically wide-spread (*Hennequin et al., 2012*), (ii) the presence of *ompK35* and/or *ompK36* mutations is prevalent in the various species of *Klebsiella* (*Rocker et al., 2020*), and (iii) a recent case study showed a single hospital had collected 87 isolates of CRE *Klebsiella* with 55% of them being a non-carbapenemase CRE phenotype (*Tamma et al., 2017a*).

## Diagnosis and treatment of non-carbapenemase CRE

To obtain the best outcome from the limited treatment options effective against CRE, a personalised approach to antibiotic dosing has been urged (*Doi, 2019*; *Doi and Paterson, 2015*; *Reyes and Nicolau, 2020*). This in turn requires rapid and accurate diagnosis. All of the currently available tests for CRE aim to identify specific carbapenemases: the Carba NP test detects OXA-48 type carbapenemases, the modified Hodge test identifies metallo-β-lactamases (*Kuchibiro et al., 2018*), and the newer more sophisticated gene-specific tests also have limitations (*Tamma et al., 2017b*; *Powell et al., 2017*; *Revez et al., 2017*; *Hong et al., 2019*; *Meunier et al., 2018*). The finding that in some environments perhaps half of all CRE cases could be caused by strains that do not encode a carbapenemase (*Tamma et al., 2017a*), and our finding that DHA-1 expression can - through epistasis with porin mutations - deliver a CRE phenotype adds a further degree of difficulty to diagnosis of CRE.

While treatment options for CRE are limited (*Reyes and Nicolau, 2020*; *Sheu et al., 2019*), this study adds benefit in two important aspects. First, combination therapy using an ESBL inhibitor such as avibactam or tazobactam could be a good option for this type of CRE (*Yahav et al., 2020*). For example, the use of ceftolozane-tazobactam combination therapy is suggested as an alternative to carbapenems in treatment of some CRE infections (*Giacobbe et al., 2018*), and a cohort study of 391 patients with ceftriaxone-resistant infections showed piperacillin-tazobactam combination therapy compared favourably with carbapenem treatment (*Harris et al., 2018*). In strains like FK688, where it is a β-lactamase such as DHA-1 that contributes to carbapenem resistance, specific inhibition of that β-lactamase with avibactam or tazobactam would increase the level of carbapenem or cephalosporin in the bacterial periplasm and thereby increase the effectiveness of drug treatment. Second, our study cautions that moves towards phage therapy being used to treat CRE should avoid the use of phages that use OmpK35 or OmpK36 as their receptor (*Rosas and Lithgow, 2022*). These phages would place selective pressure on the *Klebsiella* strain to become porin-defective since mutations to inactivate the receptor is a prime cause of phage-resistance (*Gordillo Altamirano et al., 2021*; *Gordillo Altamirano and Barr, 2021*). Phages that use OmpK35 or OmpK36 as their receptor would thereby select for porin-defects and thus inadvertently select for carbapenem-resistance; phages used therapeutically should therefore target alternate receptors (*Rosas and Lithgow, 2022*).

Finally, this study shows how the evolutionary history of a pathogenic strain can predispose for an AMR phenotype to evolve via specific genetic routes. The likelihood that antibiotic resistance will

**Table 4.** List of plasmids used in this study.

| Plasmid | Relevant characteristics* | Source/reference |
|---|---|---|
| pKD4 | Contains kanamycin resistance cassette (*kan*) flanked by FRT sites (FRT-*kan*-FRT); *oriR6K*, Amp$^R$, Km$^R$ | *Datsenko and Wanner, 2000* |
| pJET1.2/blunt | Blunt-end cloning vector for insertion of DNA fragments with single deoxyadenosine overhangs; Amp$^R$ | Thermo Scientific |
| pDonor(OmpK36) | pJET1.2/blunt carrying FRT-*kan*-FRT and *K. quasipneumoniae* FK688 *ompK36* regions (donor plasmid for lambda Red recombination-mediated repair of *ompK36* gene in FK688); Amp$^R$, Km$^R$ | This study |
| pACBSR | Arabinose-inducible promoter; I-*Sce*I endonuclease; lambda Red recombination genes, Cm$^R$ | *Herring et al., 2003* |
| pFLP-BSR | pACBSR carrying fragment length polymorphism (FLP) recombinase to excise the kanamycin cassette, temp-sensitive replication; Cm$^R$ | *Rocker et al., 2020* |
| pJP-CmR | Derivative of pJP168 for anhydrotetracycline inducible protein expression. Cm$^R$ | *Rocker et al., 2020* |
| pJP-*bla*$_{DHA-1}$ | pJP-Cm containing *bla*$_{DHA-1}$ from FK688 | This study |
| pJP-*bla*$_{OKP-B-21}$ | pJP-Cm containing *bla*$_{OKP-B-21}$ from FK688 | This study |
| pJP-*bla*$_{SHK}$ | pJP-Cm containing *CKCOFDID_01495* from FK688 | This study |
| pJP-*bla*$_{OPHC}$ | pJP-Cm containing *CKCOFDID_02113* from FK688 | This study |
| pJP-*bla*$_{DAE}$ | pJP-Cm containing *pbpG* from FK688 | This study |
| pJP-*bla*$_{TRN}$ | pJP-Cm containing *CKCOFDID_04153* from FK688 | This study |
| pJP-*bla*$_{ABH}$ | pJP-Cm containing *dhmA* from FK688 | This study |
| pJP-*bla*$_{DAC}$ | pJP-Cm containing *dacB* from FK688 | This study |

*Amp, ampicillin; Km, kanamycin; Cm, chloramphenicol.

evolve depends on the strength of selection and the availability of genetic variants that confer resistance to the antibiotic. If the genes or genetic variants that confer resistance to the antibiotic confer a decrease in fitness in environments without antibiotic, then these variants will be exceedingly rare (or absent) from the population. If antibiotic resistance genes are not supplied via horizontal gene transfer, then antibiotic resistance must evolve by the de novo mutation of a gene already present in the genome. Since the spontaneous acquisition of multiple genetic variants (for example,Δ*ompK36* and *bla*$_{DHA-1}$) is much less likely than the acquisition of a single new gene (KPC-2 carbapenemase), the evolution of non-carbapenem resistance via two loci seemed unlikely. However, if a strain already carries the *bla*$_{DHA-1}$ gene then the spontaneous evolution of a loss-of-function mutation in an extant chromosomal gene - *ompK36* - is more likely than the spontaneous acquisition of a carbapenemase gene. Exposure to ceftazidime, or any antibiotic that selects for the *bla*$_{DHA-1}$ gene, would therefore potentiate the evolution of carbapenem resistance by a single loss-of-function mutation in major porins. These results show how an individual's treatment history might shape the evolution of AMR and should be taken into consideration in order to explain the evolution of non-carbapenemase CRE.

## Materials and methods
### Chemicals and reagents
Ampicillin and tetracycline were purchased from Astral Scientific. All other antibiotics were purchased from Sigma-Aldrich in highest possible grade. A stock solution of Anhydrotetracycline (Cayman Chemical Company) in 50% ethanol was prepared to induce β-lactamase production when required.

### Bacterial strains, oligonucleotides, and cultures conditions
Plasmids, bacterial strains, and oligonucleotides used in this study are described in *Table 4*, *Table 5*, and *Table 6*, respectively. Bacterial cultures were routinely grown in Lysogeny Broth (LB) or cation-adjusted Mueller-Hinton Broth (CAMHB) media at 37°C with shaking at 200 rpm, unless otherwise stated. When required, antibiotics used for the selection of antibiotic resistance markers were supplemented in growth media at the following concentrations: ampicillin 100 μg/mL; kanamycin 30 μg/mL;

**Table 5.** List of strains used in this study.

| Strain | Relevant characteristics* | Source or reference |
|---|---|---|
| *K. quasipneumoniae* | | |
| FK688 Δ*ompK36* pNAR1 | Wildtype, clinical isolate from a bloodstream infection case from the First Affiliated Hospital of Wenzhou Medical University, China. Expresses β-lactamase $bla_{OKP-B-21}$. Deficient in *ompK35* and *ompK36* porin expression. Harbours a 258 kb plasmid pNAR1 (Amp$^R$, Tet$^R$, Rif$^R$, Trp$^R$, Stp$^R$, Ery$^R$, Sdz$^R$, Cip$^R$). | *Bi et al., 2017* |
| Δ*ompK36* pNAR1Δ*bla*$_{DHA-1}$ | FK688 with a 17 kb deletion from *tnpA-sul1* in pNAR1. | This study |
| Δ*ompK36* pNAR1$^-$ | FK688 cured of pNAR1. | This study |
| *ompK36*$^+$ pNAR1 | FK688 with a genetically repaired and functional *ompK36* gene. Carries pNAR1. | This study |
| *ompK36*$^+$ pNAR1Δ*bla*$_{DHA-1}$ | FK688 with a genetically repaired and functional *ompK36* gene. It has a 17 kb deletion from *tnpA-sul1* in pNAR1. | This study |
| FK688-GFP$^+$ | FK688 with a constitutively expressed green fluorescent protein (GPF). GFP gene inserted downstream of the *glmS* gene via pGRG-eGFP. | This study |
| A2(o) Δ*ompK36* pNAR1Δ*bla*$_{DHA-1}$ | Evolved strain from Kq1. It has a 17 kb deletion from *tnpA-sul1* in pNAR1. Forms opaque colonies. | This study |
| A2(t) Δ*ompK36* pNAR1Δ*bla*$_{DHA-1}$ | Evolved strain from Kq1. It has a 17 kb deletion from *tnpA-sul1* in pNAR1. Forms translucent colonies. | This study |
| B3(o) *ompK36*$^+$ pNAR1Δ*bla*$_{DHA-1}$ | Evolved strain from Kq4. It has a 17 kb deletion from *tnpA-sul1* in pNAR1. Forms opaque colonies. | This study |
| B3(t) *ompK36*$^+$ pNAR1Δ*bla*$_{DHA-1}$ | Evolved strain from Kq4. It has a 17 kb deletion from *tnpA-sul1* in pNAR1. Forms translucent colonies. | This study |
| *K. pneumoniae* | | |
| B5055 | Hypermucoviscous phenotype. Wildtype, clinical isolate, serotype K2;O1 | Statens Serum Institut, Denmark |
| B5055 nm | B5055 deletion mutant Δwza-wzc::km (non-mucoid); Km$^R$ | Prof. Richard Strugnell University of Melbourne |
| *E. coli* | | |
| DH5α | F$^-$ endA1 *hsd*R17(r$_K^-$, m$_K^+$) *sup*E44 *λ* – *thi*-1 *rec*A1 *gyr*A96 *rel*A1 *deo*R Δ(*lac*ZYA-*arg*F) U169 Φ80*dlac*ZΔM15; Nal$^R$ *E. coli* DH5α was used for cloning purposes | Invitrogen |
| ATCC 25922 | CLSI control strain for antimicrobial susceptibility testing | ATCC |
| BW25113 (WT) | *rrn*B3 Δ*lac*Z4787(::rrnB-3) *hsd*R514 Δ(*ara*D-*ara*B)567 Δ(*rha*D-*rha*B)568, *rph*-1 | *Baba et al., 2006* |

*Amp, ampicillin; Tet, tetracycline; Rif, rifamycin; Trp, trimethoprim; Stp, streptomycin; Ery, erythromycin; Sdz, sulfadiazine; Cip, ciprofloxacin; Nal, nalidixic acid.

chloramphenicol 34 µg/mL; ceftazidime: 0.125 µg/mL, 0.25 µg/mL, 0.5 µg/mL, 1 µg/mL, or 2 µg/mL; imipenem: 0.03 µg/mL, 0.06 µg/mL, 0.125 µg/mL, 0.25 µg/mL, or 0.5 µg/mL.

## Genome sequencing and evaluation

gDNA of the *Klebsiella* isolates was prepared from solid media scrapings of pure culture using the GenElute Bacterial Genomic DNA Kit (Sigma-Aldrich) and the Gram-negative bacteria protocol. High molecular weight DNA was then isolated using a 0.6× ratio of sample (200 µLl) to AMPure XP-beads (120 µL; A63882, Beckman Coulter). gDNA was sequenced in parallel on the Oxford Nanopore GridION and Illumina Nextseq 500. High molecular weight DNA was prepared as a Nanopore sequencing library, according to the manufacturer's protocols using a ligation sequencing kit (SQK-LSK109, Oxford Nanopore), with minor modifications. All mixing steps for the DNA sample were done by gently flicking the microfuge tube instead of pipetting, and the optional shearing step was omitted.

DNA repair treatment was carried out using NEBNext FFPE DNA Repair Mix (M6630, New England Biolabs). End repair and A-tailing were performed with NEBNext Ultra II End Repair/dA-tailing Module

**Table 6.** List of oligonucleotide primers used in this study.

| Primer | Sequence (5–3')* | Description |
|---|---|---|
| **Construction of FK688 OmpK36⁺ strains** | | |
| K36_insert-R | gcgcgacctactacttcaacaaaaacatgtccacctatgttgactacaaaatcaacctgctg | |
| K36_insert-F | gttgaagtagtaggtcgcgcccacgtcaacatatttcaggatgtcctggtcgcc | |
| K36_Km-F | ctaaggaggatattcatatggtcgcaagctgcataacaaa | |
| K36_Km-R | gaagcagctccagcctacacattagaactggtaaaccaggcccag | |
| K36_ISceI-R | tagggataacagggtaatgcccgacggtgatatccatc | |
| K36_ISceI-F | tagggataacagggtaatgcttcggtacctctgtaacttatga | Construction of pDonor(OmpK36) plasmid |
| pKD4-F | tgtgtaggctggagctgcttc | |
| pKD4-R | catatgaatatcctccttag | Kanamycin cassette from pKD4 |
| **Cloning of putative β-lactamases genes for anhydrotetracycline-inducible expression** | | |
| blaDHA-1_For_NR | gtccCCATGGtgaaaaaatcgttatctgcaac | |
| blaDHA-1_Rev_NR | cgtcAAGCTTattccagtgcactcaaa | |
| blaOKP_F_NR | tagcGAATTCatgcgttatgttcgcctgtgcc | |
| blaOKP_R_NR | gcatAAGCTTctagcgctgccagtg | |
| blaSHK1_F_NR | gttcCCATGGtgataagaaaaccactggcc | |
| blaSHK1_R_NR | atgcAAGCTTaacgcagctcgcg | |
| blaOPHC2_F_NR | ctagGAATTCatgacaccagctccctttttataccctgac | |
| blaOPHC2_R_NR | acggAAGCTTtcgctgtgatcggtgtt | |
| blaDAE1_F_NR | tgcaGAATTCatgatgccgaaatttcgagtctctttgc | |
| blaDAE1_R_NR | gatcAAGCTTtttaatcgttctgcgcg | |
| blaABH1_F_NR | acgtCCATGGTGaacagattatccctgatcc | |
| blaABH1_R_NR | gatcAAGCTTacaaccgatcggcg | |
| blaDAC1_F_NR | aaggCCATGGtgcgatttcccagatttatc | |
| blaDAC1_R_NR | aagcAAGCTTtagttgttctggtacaaatcc | |
| blaTRN1_F_NR | cgtaCCATGGtgactgaacgggtttattacac | |
| blaTRN1_R_NR | aatcAAGCTTacgtcagggaatagctgatc | |
| pJPMCS_For | cctaattttttgttgacactctatcattg | |
| pJPMCS_Rev | gccaggcaaattctgttttatcagaccg | pJP-CmR-gene insert sequencing primers |

*Restriction endonuclease recognition sites are capitalised.

(E7546, New England Biolabs), and the sample was incubated at 20°C for 5 min and then 65°C for 5 min. End-repaired product was purified with 1×Agencourt AMPure XP beads.

Adapters provided in the respective library kits were ligated to the DNA with Quick T4 DNA Ligase (M2200L, New England Biolabs), and samples were incubated at room temperature for 10 min. Purification and loading of adapted libraries on an appropriate flow cell (R9.4.1, FLO-MIN106D, Oxford Nanopore) were completed as stated in the manufacturer's protocol and sequenced using the appropriate MinKNOW workflow. The library was base-called using Guppy (ont-guppy-for-gridion, 3.0.6). Reads with a length less than 1000 bp were discarded. The sample had an estimated Nanopore coverage of 884-fold, with an average read length of 7445 bp and a range of 1000–191,716 bp.

Illumina sequences were prepared on a NextSeq 500 platform, with 150 bp paired-end chemistry. Reads were trimmed to remove adaptor sequences and low-quality bases with Trimmomatic (*Bolger et al., 2015*), with Kraken used to investigate contamination ( *Wood and Lu, 2015*, v0.10.5-beta).

Assembly involved long-read-only assembly of long reads, followed by short-read correction. In brief, Nanopore reads were downsampled using Filtlong (*Wick and Menzel, 2018*, v0.2.0) to retain the highest quality reads (10% of all, equivalent to an estimated 88-fold coverage). These reads were assembled using Canu (*Koren S et al., 2018*, v1.8), with an expected genome size of 5,000,000 bp. The assembled contigs output by Canu were circularised where appropriate and validated through a read mapping approach in Geneious Prime (2019.2.1) before short-read correction. The corrected assembly was oriented to *dnaA* and annotated using Prokka (*Seemann, 2020*). The sample had varied coverage across assembled molecules, ranging in Illumina coverage between 156-fold for the chromosome and 188-fold for the plasmid.

## Plasmid annotation

Prokka v1.14.0 was employed to predict pNAR1 genes. The translated gene sequences were used to search against NCBI nr and CARD resistance databases (Comprehensive Antibiotic Resistance Database) with the blastp algorithm and Resistance Gene Identifier software (*Alcock et al., 2020*), respectively (e value≤$10^{-5}$). The predicted genes received an annotation file containing credible resistance genes (cut-off as "Perfect" or "Strict") and putative resistance genes (cut-off as "Loose").

## Membrane protein analysis

### Membrane purification and isolation

Overnight cultures of strains were diluted 1:100 in 200 mL CAMHB and grown until $OD_{600}$ = ~0.5. Cells were harvested by centrifugation (10,000 × *g*, 10 min, 4°C) and resuspended in 10 mM Tris-HCl, pH 7.5. The centrifugation was repeated, and cells resuspended in Tris-Sucrose buffer (10 mM Tris-HCl, pH 7.5, 0.75 M sucrose). Peptidoglycan was degraded at a final concentration of 50 µg/mL lysozyme, and host serine proteases were inhibited at a final concentration of 2 mM phenylmethylsulfonyl fluoride. The outer membrane was destabilised for lysis in two volumes of 1.65 mM EDTA, pH 7.5. Cells were incubated on ice for 10 min and then lysed using an AVESTIN Emulsiflex-C3 (4 passes at ~15,000 psi). Cell lysates were centrifuged (15,000 × *g*, 20 min, 4°C) to remove cell debris. The supernatant was collected, and total membranes were pelleted by ultracentrifugation (132,000 × *g*, 45 min, 4°C) using a 70.1 Ti rotor. Membrane pellets were resuspended and pooled from duplicate samples in ~8 mL TES buffer (2.2 mM Tris-HCl, pH 7.5, 1.1 mM EDTA, and 0.25 M sucrose) and ultracentrifuged as before. Membrane pellets were resuspended in 200 µL of 25% sucrose in 5 mM EDTA, pH 7.5, and stored at −80°C.

### Protein expression and analysis

Total (outer and inner) membranes from *K. quasipneumoniae* were purified following the method of *Dunstan et al., 2017* with minor modifications. Membrane proteins were quantified with a NanoDrop 1000 Spectrophotometer (Thermo Scientific) and standardised to equivalent concentrations. Samples (~2 µg) were loaded onto a sSDS-PAGE containing 11% (wt/vol) 37.5:1 acrylamide-bisacrylamide, 0.375 M Tris (pH 8.8), 0.1% (wt/vol) SDS, and 0.5 mM EDTA in the separating gel and 4% (wt/vol) 37.5:1 acrylamaide-bisacrylamide, 0.25 M Tris (pH 6.8), 0.1% (wt/vol) SDS, and 0.5 mM EDTA in the stacking gel.

Proteins (~2 µg) were transferred to a 0.2 µm nitrocellulose membrane (Bio-Rad) using a Trans-Blot Turbo Transfer System (Bio-Rad; 0.6 A, 25 V, 15 min), which was blocked overnight in TBST (tris-buffered saline; 0.1% Tween 20) containing 5% skim milk.

OmpK36 was detected by western blot using polyclonal antibodies raised in rabbits against the *E. coli* homolog OmpC, which is cross-reactive to OmpK35 and OmpK36 in *Klebsiella*. For use as a loading control, the outer-membrane protein BamD was detected using a α-BamD antibody raised in rabbits against the *E. coli* BamD homolog. Goat Anti-Rabbit IgG antibody, HRP-conjugate (Sigma-Aldrich) was used as the secondary antibody. All antibodies were used at a 1:20,000 dilution in TBST containing 2% skim milk. Proteins were detected by chemiluminescence.

## MIC determination

Antimicrobial susceptibility testing was performed by broth microdilution method using CAMHB according to the guidelines the Clinical and Laboratory Standards Institute (CLSI) M07-10th Ed. Document *CLSI, 2015*. The resistance of antimicrobial agents was interpreted according to the criteria of

*CLSI, 2022*. The assays were performed in biological triplicate with at least two technical replicates. *E. coli* ATCC 25922 was used as a quality-control strain.

## Mutant construction

A *K. quasipneumoniae* FK688 strain containing a repaired *ompK36* gene (*ompK36⁺*) was constructed using the "gene gorging" technique (*Herring et al., 2003*; *Cherepanov and Wackernagel, 1995*; *Lee et al., 2009*; *Datsenko and Wanner, 2000*). A donor plasmid was made that contained the repaired *ompK36* gene upstream of a kanamycin-resistant cassette and ~500 bp of FK688 genomic region downstream of the *ompK36* gene and flaked by I-*Sce*I- endonuclease recognition sites. The PCR products were gel purified, cloned into pJET1.2/blunt, and confirmed by sequencing. The primers used are listed in *Table 6*. The donor plasmid and pACBSR carrying L-arabinose-inducible I-*Sce*I endonuclease and $\lambda$-Red recombinase genes were transformed into FK688 by electroporation. Co-transformants were inoculated into LB containing chloramphenicol and 0.2% (*w/v*) L-arabinose and incubated overnight at 30°C with shaking. Engineered strains were isolated on LB-agar containing kanamycin and cured of the donor and pACBSR plasmids (by their sensitivity to chloramphenicol), and mutants were confirmed by PCR. The self-curing plasmid pFLP-BSR was then used to excise the kanamycin cassette (*Cherepanov and Wackernagel, 1995*).

## Validation testing of candidate β-lactamases

The coding sequences of β-lactamase and DeepBL candidate genes were amplified from FK688 gDNA using Fusion High-Fidelity DNA polymerase (New England BioLabs) with the oligonucleotide primers listed in *Table 6*. The PCR products and the anhydrotetracycline (ATc)-inducible expression vector pJP-CmR were digested with restriction enzymes using either *Nco*I and *Hin*dIII, or *Eco*RI and *Hin*dIII (New England BioLabs) and ligated to create the plasmids listed in *Table 5*. Plasmids were verified by sequencing, transformed into *E. coli* BW25113 or *K. quasipneumoniae* FK688, and selected with chloramphenicol. Target gene expression was induced with 35 ng/mL ATc. The parental plasmid pJP-CmR was used as the control in all experiments.

## Plasmid maintenance assessment

A mutation accumulation experiment was performed to evaluate pNAR1 stability in the FK688 Δ*ompK36* pNAR1 strain. The strain was cultured on LB agar (no antibiotics) from a glycerol stock, corresponding to passage #1 (P1). From this plate, 10 colonies were replica cultured on LB agar containing 10 µg/mL ceftazidime (LB-CAZ) and LB agar without antibiotics (LB-only). Five colonies from P1 were individually subcultured on five LB agar plates without antibiotics (P2). One colony from each P2 plate was similarly passaged to P20, with replica plating of 10 colonies on LB-CAZ and LB-only after each passage. Therefore, 50 colonies were screened after each passage. Colonies that grew on LB-only but not on LB-CAZ (CAZS) were assessed for pNAR1 maintenance by PCR.

## Fitness assays evaluation

Competitive fitness assays of strains relative to a GFP-expressing reference *K. quasipneumoniae* FK688-GFP⁺ strain were performed as described in *Barber et al., 2021* with some modifications. Single colonies of ancestral, evolved, and reference strains were grown overnight at 37°C in 3 mL LB media (with and without antibiotic selection) with shaking in separate 15 mL falcon tubes. At saturation, the strain of interest and reference strain were mixed (100 µL:100 µL), diluted in PBS, and measured by fluorescence-activated cell sorting to determine the unadjusted proportions of the two strains. Based off these unadjusted values, volumes of experimental and reference strains were then modified to create a 1:1 cell density ratio, which is the initial starting frequency. The mixture of strains was then diluted 1:1000 (3 µL in 3 mL of LB) before propagating into fresh LB media each day (10 generations per day). 500 µL of sample was taken each day, diluted in 1×PBS (PBS [pH 7.4]: 137 mM NaCl, 2.7 mM KCl, 10 mM Na₂HPO₄, and 2 mM KH₂PO₄), and measured by flow cytometry (LSR Fortessa X20a) for the proportion of experimental to reference strains. A maximum total count of 50,000 events was used. The selection coefficient (S) per generation for each experimental strain relative to the reference strain was calculated by taking the natural logarithm of the ratio of experimental to reference strains at the initial and the final time point and dividing by the number of generations passed (*McDonald, 2019*) as described by the following regression model formula:

$$S = \frac{\ln\left(\dfrac{E_t}{R_t}\right) - \ln\left(\dfrac{E_0}{R_0}\right)}{T}$$

Where T=time (generations); E=frequency of evolved strain; *R*=frequency of the reference (GFP-labelled) strain; $E_t$, $R_t$ = frequencies at time "t"; $E_0$, $R_0$=initial frequencies.

In other words, when S=0, the strains are equally fit; when S is positive, the evolved strain is more fit than the reference strain; and when S is negative, the evolved strain is less fit than the reference strain.

The relative fitness of the GFP-expressing FK688 strain is on average 0.040 for all experiments compared it to the ancestral FK688: Δ*ompK36* pNAR1strain with no GFP expression. The final selection coefficient of all graphs is calculated after normalising the relative fitness of the FK688-GFP$^+$ against the ancestor FK688.

## Evolution experiments

To set up the evolution experiment, single clones of the drug-resistant, non-functional porin (Δ*ompK36* pNAR1) strain and its homologous, membrane engineered (*ompK36*$^+$pNAR1) strain were used to seed 20 replicates each in 3 mL LB media and serially passaged in antibiotic-free media for 24 hr at 37°C with shaking in separate 15 mL falcon tubes. Every 24 hr, the populations were diluted 1:1000 (3 μL in 3 mL of LB), equating to roughly 10 generations per day. The strain populations were evolved for approximately 200 generations. The relative growth rate of all lineages and the parental strains was measured every 50 generations using fitness assays, and colony morphological features were assessed throughout the experiment. After 200 generations, four single clones of evolved strains with different morphological characteristics were selected for whole genome sequencing. After every 20 and 50 generations, 500 μL of evolved strains were mixed with 500 μL of 50% glycerol and stored at –80°C.

## Growth curves

Overnight bacterial cultures were subcultured 1:100 in CAMHB and grown until mid-log phase (OD$_{600}$ of 0.6–0.8). Cells were then diluted in CAMHB to an OD$_{600}$ of 0.05. Three biological replicates (each in technical replicates) were grown in a 96-well plate using the Tecan Spark 10 M. The plate was enclosed in a hydration chamber for 24 hr at 37°C with orbital shaking (200 rpm) amplitude 3 mm, and cell density (OD$_{600}$) measurements were recorded every 60 min.

## Capsule polysaccharide analysis

Capsular polysaccharides of *Klebsiella* strains were extracted by the phenol-extraction method (*Campos et al., 2004*) and quantified using colorimetric assays for glucuronic acid as previously described (*Blumenkrantz and Asboe-Hansen, 1973*). Capsule extraction: overnight bacterial cultures were subcultured 1:50 in LB media and grown until the mid-log phase (OD600~0.4–0.6). Cells were collected by centrifugation (5000 × *g*, 15 min, 5°C), and the pellet was resuspended in 1 mL of dH2O. After centrifugation (14,000 × *g*, 10 min), the pellet was resuspended in 500 μl dH2O, and viable counts were determined. Samples were incubated at 68°C for 2 min, and 500 μL phenol (Sigma #P4557) was added and mixed by inversion. Following incubation at 68°C for 30 min, the mixture was cooled on ice for 2 min. 500 μL chloroform (Sigma-Aldrich #472476) was added, and the mixture was inverted. Samples were centrifuged (10,000 × *g*, 5 min), and approximately 400 μL of cell-free supernatant was mixed with 1 ml absolute ethanol, incubated at –20°C for 20 min, and washed with 70% ethanol. The carbohydrate-containing precipitate was resuspended in 500 μL dH2O. Samples were stored at 4°C. 200 μL of capsular material was mixed with 1.2 mL of 12.5 mM sodium tetraborate (Borax; NaBH4) in concentrated H$_2$SO$_4$. The mixtures were vigorously vortexed, boiled for 5 min, and then cooled on ice for 10 min before the addition of 20 μL of 0.15% 3-hydroxydiphenol in 0.5% NaOH. Absorbance was measured at 520 nm. A duplicate sample from each strain was prepared as described above but treated with 0.5% sodium hydroxide alone and used to measure the background absorbance at 520 nm. The glucuronic acid concentration in background-subtracted values for each sample was determined from a standard curve of $_D$-Glucuronic acid.

## Phylogeny

377 complete *Klebsiella* genomes from the NCBI database were used to make a global phylogeny (downloaded May 2020). Roary (3.11.2; *Page et al., 2015*) was used to align the genomes and extract

the core genomes. The core genomes were used to generate a phylogenetical tree using RAxML v8.2.12 (*Stamatakis, 2014*) with a general time reversible nucleotide substitution model with rate heterogeneity modelled with a gamma distribution (GTR +GAMMA). Branch supports were estimated using 1000 bootstrap replicates. Kleborate v0.3.0 (*Wyres et al., 2016*) was used to identify MLST types. Perfect sequence matches of 7 MLST loci are indicated as ST and single- or double-locus variant of the exact ST is represented as STxx-1LV or STxx-2LV, respectively. Visualisation of the tree was generated with R using the package ggtree.

## Acknowledgements

This work was supported by the National Health and Medical Research Council (grant number 1092262) and a Monash BDI Postgraduate Research Scholarship.

## Additional information

### Funding

| Funder | Grant reference number | Author |
|---|---|---|
| National Health and Medical Research Council | 1092262 | Trevor Lithgow |

The funders had no role in study design, data collection and interpretation, or the decision to submit the work for publication.

### Author contributions

Natalia C Rosas, Conceptualization, Data curation, Investigation, Visualization, Methodology, Writing – original draft, Writing – review and editing; Jonathan Wilksch, Jake Barber, Jiahui Li, Yanan Wang, Zhewei Sun, Andrea Rocker, Chaille T Webb, Laura Perlaza-Jiménez, Christopher J Stubenrauch, Vijaykrishna Dhanasekaran, Jiangning Song, George Taiaroa, Mark Davies, Richard A Strugnell, Qiyu Bao, Tieli Zhou, Investigation; Michael J McDonald, Conceptualization, Supervision, Investigation, Writing – original draft, Writing – review and editing; Trevor Lithgow, Conceptualization, Resources, Formal analysis, Supervision, Funding acquisition, Investigation, Writing – original draft, Project administration, Writing – review and editing

### Author ORCIDs

Natalia C Rosas ⓘ http://orcid.org/0000-0003-2343-2394
Jake Barber ⓘ http://orcid.org/0000-0001-9279-7763
Christopher J Stubenrauch ⓘ http://orcid.org/0000-0003-4388-3184
Vijaykrishna Dhanasekaran ⓘ http://orcid.org/0000-0003-3293-6279
Michael J McDonald ⓘ http://orcid.org/0000-0002-5735-960X

### Decision letter and Author response

Decision letter https://doi.org/10.7554/eLife.83107.sa1
Author response https://doi.org/10.7554/eLife.83107.sa2

## Additional files

### Supplementary files
• MDAR checklist

### Data availability

Complete genome sequence data for *Klebsiella quasipneumoniae* subsp. *similipneumoniae* FK688 has been deposited at the NCBI (Accessions: CP072505, CP072506 and CP072507) and annotated through the Prokaryotic Genome Annotation Pipeline (PGAP) under the Bioproject PRJNA717371 and Biosample SAMN18498882.

The following dataset was generated:

| Author(s) | Year | Dataset title | Dataset URL | Database and Identifier |
|---|---|---|---|---|
| Rosas NC, Wilksch JJ, Barber J | 2021 | The evolutionary mechanism of complex carbapenem-resistant phenotypes in Klebsiella spp | https://www.ncbi.nlm.nih.gov/bioproject/PRJNA717371 | NCBI BioProject, PRJNA717371 |

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
