## [Editor Report]

In this study, the authors examine the mechanisms of resistance to carbapenem in *Klebsiella* quasipneumoniae, through a non–carbapenemase mechanism. The evidence – which includes a combination of experimental and computational approaches – is compelling. It offers a set of findings that make a very important contribution to our understanding of not only how antimicrobial resistance evolves, but how to integrate methods and data of different kinds towards understanding a complex evolutionary phenomenon.

---

## [Decision Letter]

**Decision letter after peer review:**

Thank you for submitting your article "Treatment history shapes the evolution of complex carbapenem–resistant phenotypes in *Klebsiella* spp." for consideration by *eLife*. Your article has been reviewed by 2 peer reviewers, and the evaluation has been overseen by a Reviewing Editor and George Perry as the Senior Editor. The following individual involved in review of your submission has agreed to reveal their identity: Rohan Maddamsetti (Reviewer #2).

Essential revisions:

You must either provide data on the history of the patient, or substantially reframe the claims made. Currently, while the paper has compelling data and analyses, the central claims are not supported. In general, toning down some of the claims, and making them much more consistent with the data and results (which are strong) are needed.

*Reviewer #1 (Recommendations for the authors):*

The manuscript was very interesting and had a good flow. Apart of the problem with the experimental evolution (Figure 5, lines 193–215):

– Figure 1B: Please enlarge the font size of the colour key.

– Figure 1B: What do you mean by: "The outer ring colours indicates the distribution of the sequence type classifications"? Where is the colour code?

– line 138: "…resistance to we used…" should be "…resistance, we used…".

– Figure 3B: Please indicate the region enlarged in 3C.

– Line 156: Why figure S4?

– Line 193: What is the phenotype of the ompK36+ strain (before the evolution experiment)?

– Line 198: What is the B3 population?

– Line 521: In this experiment you assume that the GFP expression has no effect in our conditions. His is not necessarily true. You can validate this by checking whether you get S=0 when you compare the non–evolved strain with no GFP expression to the one with GFP expression. If this was already done, it is not clear from the text.

– Line 293: Figure 6E does not exist.

– Table S7 and S8: The actual location of the mutations is not shown. Please provide the details.

[Editors' note: further revisions were suggested prior to acceptance, as described below.]

Thank you for resubmitting your work entitled "The evolutionary mechanism of complex carbapenem–resistant phenotypes in *Klebsiella* spp." for further consideration by *eLife*. Your revised article has been evaluated by George Perry (Senior Editor) and a Reviewing Editor.

The manuscript has been improved but there are some remaining issues that need to be addressed, as outlined below:

Please pay close attention to the recommendations from reviewer #2, which include suggestions regarding the use of technical jargon and the use of precise language. It is very important to not further obfuscate genetics understanding which is already confusing to many.

*Reviewer #2 (Recommendations for the authors):*

This paper is a valuable contribution to the field of AMR evolution. I have one comment.

The authors did an admirable job of dissecting the genetic basis of carbapenem resistance. Given the "simple" genetic basis (simple as in explained by 2 loci), I suggest cutting language about "complex" genetic basis or "omnigenic inheritance" because these terms have a specific technical meaning in genetics. "complex traits" are those that are generated by many (hundreds of) loci, and "omnigenic" *specifically* refers to traits in which most of the genome contributes. See the original paper: https://www.ncbi.nlm.nih.gov/pmc/articles/PMC5536862/

Along these lines, perhaps a title like: "*Klebsiella* spp. can evolve carbapenem resistance without carbapenemases"? Something that conveys the main point more precisely.

Lines 78–80: Cut "Given the omnigenic nature of this type of AMR phenotype" so that the sentence is: "Understanding the details of these evolutionary forces could provide a means to re–sensitize populations of bacteria to carbapenems."

Line 90: cut "complex basis again so: "explain the evolution of non–carbepenemase CRE."

---

## [Author Response]

Essential revisions:You must either provide data on the history of the patient, or substantially reframe the claims made. Currently, while the paper has compelling data and analyses, the central claims are not supported. In general, toning down some of the claims, and making them much more consistent with the data and results (which are strong) are needed.

We appreciate the point and have substantially reframed the claims with regard to patient history. This revision is detailed under each of the points made by Reviewer #1.

Reviewer #1 (Recommendations for the authors):The manuscript was very interesting and had a good flow. Apart of the problem with the experimental evolution (Figure 5, lines 193–215):– Figure 1B: Please enlarge the font size of the colour key.– Figure 1B: What do you mean by: "The outer ring colours indicates the distribution of the sequence type classifications"? Where is the colour code?

We have made these corrections and have revised the Legend for Figure 1B to clarify:

“(B) Maximum likelihood phylogenetic tree of 377 publicly available *Klebsiella* genomes shows *K. pneumoniae* and *K. quasipneumoniae* as distinct species. The inner ring colors refer to the country of isolation according to the key and further data is described in Figure supplement 1 and Figure 1- source data 1.”

– line 138: "…resistance to we used…" should be "…resistance, we used…".

Corrected (now line 136).

– Figure 3B: Please indicate the region enlarged in 3C.

We have revised Figure 3B to indicate the region enlarged in Figure 3C with a black arc line, and clarified the Legend in this regard.

– Line 156: Why figure S4?

Corrected, we have move the figure reference to make it clear why we reference this figure (now Figure 3—figure supplement 2) now Line 152.

– Line 193: What is the phenotype of the ompK36+ strain (before the evolution experiment)?

The ompK36^+^, pNAR1 strain (Lineage B, generation 0) is resistant to penicillin and cephem antibiotics, but susceptible to carbapenem antibiotics. We tested the antimicrobial susceptibility profile before the evolution experiment (Table 1) and compared these values with the evolved strains (Table 3).

– Line 198: What is the B3 population?

We appreciate the point regarding the nomenclature and have revised Figure 5 to include a graphic (Figure 5A) depicting the lineages (A or B, denoting OmpK36 status) and populations (20 populations were evolved in parallel). Population B3 is one of the 20 populations that evolved from lineage B, the ompK36+ lineage.

We have revised the RESULTS text (Lines 184 onward), citing Figure 5A and explaining which strains are referred to as A2 and B3 (sentence starting Line 189).

– Line 521: In this experiment you assume that the GFP expression has no effect in our conditions. His is not necessarily true. You can validate this by checking whether you get S=0 when you compare the non–evolved strain with no GFP expression to the one with GFP expression. If this was already done, it is not clear from the text.

Thank you for pointing out this out. We have now explained more clearly what we have done in the Methods section.

We do account for the cost of GFP. We do this by first comparing the ancestor strain (FK688 ∆ompK36pNAR1) to the ancestor-GFP strain and then setting that as fitness = zero. Then, we compared every strain that we are interested into to the GFP strain, and report the fitness of each strain relative to the ancestor. In this way we can obtain information of the relative fitness of each strain, even if the GFP does have a fitness effect.

– Line 293: Figure 6E does not exist.

Corrected.

– Table S7 and S8: The actual location of the mutations is not shown. Please provide the details.

Table S7 and S8 (Figure 5—figure supplement 1 and 2) have been revised to show the position of each SNP and genome modification.

[Editors' note: further revisions were suggested prior to acceptance, as described below.]

The manuscript has been improved but there are some remaining issues that need to be addressed, as outlined below:Please pay close attention to the recommendations from reviewer #2, which include suggestions regarding the use of technical jargon and the use of precise language. It is very important to not further obfuscate genetics understanding which is already confusing to many.Reviewer #2 (Recommendations for the authors):This paper is a valuable contribution to the field of AMR evolution. I have one comment.The authors did an admirable job of dissecting the genetic basis of carbapenem resistance. Given the "simple" genetic basis (simple as in explained by 2 loci), I suggest cutting language about "complex" genetic basis or "omnigenic inheritance" because these terms have a specific technical meaning in genetics. "complex traits" are those that are generated by many (hundreds of) loci, and "omnigenic" *specifically* refers to traits in which most of the genome contributes. See the original paper: https://www.ncbi.nlm.nih.gov/pmc/articles/PMC5536862/

Thanks, and this is very reasonable request. To address the reviewer’s concerns, we have remove the word “complex” from the abstract (Line 37), so it now reads:

“…could explain the genetic basis of carbapenem-resistance found in many enteric-pathogens.”

There are no other instances of the word complex or omnigenic in the manuscript.

Along these lines, perhaps a title like: "Klebsiella spp. can evolve carbapenem resistance without carbapenemases"? Something that conveys the main point more precisely.

We have changed the title to incorporate the spirit of the reviewer’s recommendation, removing the word “complex” and including “non-carbapenemase”. The new title:

“The evolutionary mechanism of non-carbapenemase carbapenem-resistant phenotypes in *Klebsiella* spp.”

Lines 78–80: Cut "Given the omnigenic nature of this type of AMR phenotype" so that the sentence is: "Understanding the details of these evolutionary forces could provide a means to re–sensitize populations of bacteria to carbapenems."

This has been done (now Line 79), the manuscript now reads:

“Understanding the details of these evolutionary forces could provide a means to re-sensitize populations of bacteria to carbapenems.”

Line 90: cut "complex basis again so: "explain the evolution of non–carbepenemase CRE."

We have done this, now Line 90:

“…which may explain the evolution of non-carbepenemase CRE.”